# Cytoglobin regulates NO-dependent cilia motility and organ laterality during development

Elizabeth R. Rochon[1], Jianmin Xue[2], Manush Sayd Mohammed[3], Caroline Smith[2], Anders Hay-Schmidt[4], Anthony W. DeMartino[1], Adam Clark[1], Qinzi Xu[1], Cecilia W. Lo[5], Michael Tsang[3], Jesus Tejero[2,6,7,8], Mark T. Gladwin[1]✉ & Paola Corti[9]✉

Cytoglobin is a heme protein with unresolved physiological function. Genetic deletion of zebrafish cytoglobin (*cygb2*) causes developmental defects in left-right cardiac determination, which in humans is associated with defects in ciliary function and low airway epithelial nitric oxide production. Here we show that Cygb2 co-localizes with cilia and with the nitric oxide synthase Nos2b in the zebrafish Kupffer's vesicle, and that cilia structure and function are disrupted in *cygb2* mutants. Abnormal ciliary function and organ laterality defects are phenocopied by depletion of *nos2b* and of *gucy1a*, the soluble guanylate cyclase homolog in fish. The defects are rescued by exposing *cygb2* mutant embryos to a nitric oxide donor or a soluble guanylate cyclase stimulator, or with over-expression of *nos2b*. Cytoglobin knockout mice also show impaired airway epithelial cilia structure and reduced nitric oxide levels. Altogether, our data suggest that cytoglobin is a positive regulator of a signaling axis composed of nitric oxide synthase–soluble guanylate cyclase––cyclic GMP that is necessary for normal cilia motility and left-right patterning.

Cytoglobin (Cygb) is a heme-containing globin with a hexacoordinate heme moiety expressed at low micromolar concentrations, suggesting that unlike its pentacoordinate relatives, hemoglobin and myoglobin, Cygb may not be involved in oxygen transport or storage. Instead, through electron transfer reactions, Cygb may control nitric oxide (NO) metabolism through its redox status[1–4]. The reduced deoxygenated heme participates in electron and proton transfer reactions to reduce nitrite to bioactive NO[5–7], and the oxygen bound heme reacts with NO to oxidize the NO to inert nitrate[8–11], thus participating in oxygen dependent NO homeostasis. Studies in mouse knockout models suggest roles for Cygb in regulating vascular tone through NO dioxygenation (scavenging)[11] and in protecting the cells from superoxide toxicity through a superoxide dismutase (SOD) function[12]. Our group and others have found that Cygb is a redox regulated hemoprotein, with the sequential oxidation of two surface cysteines increasing the iron-histidine affinity and the open probability of the heme pocket for ligand binding and nitrite reduction[4,7,13]. Furthermore, oxidized ferric Cygb is reduced by the cytochrome $b_5$ reductase

[1]Department of Medicine, University of Maryland School of Medicine, Baltimore, MD 21201, USA. [2]Pittsburgh Heart, Lung, and Blood Vascular Medicine Institute, Department of Medicine, University of Pittsburgh School of Medicine, Pittsburgh, PA 15213, USA. [3]Department of Developmental Biology, University of Pittsburgh School of Medicine, Pittsburgh, PA 15260, USA. [4]Department of Odontology, Faculty of Health and Medical Sciences, University of Copenhagen, Copenhagen, Denmark. [5]Department of Developmental Biology, Rangos Research Center, University of Pittsburgh School of Medicine, Pittsburgh, PA 15201, USA. [6]Division of Pulmonary, Allergy and Critical Care Medicine, University of Pittsburgh School of Medicine, Pittsburgh, PA 15261, USA. [7]Department of Bioengineering, University of Pittsburgh Swanson School of Engineering, Pittsburgh, PA 15260, USA. [8]Department of Pharmacology and Chemical Biology, University of Pittsburgh, Pittsburgh, PA 15261, USA. [9]Department of Biochemistry and Molecular Biology, University of Maryland School of Medicine, Baltimore, MD 21201, USA. ✉e-mail: MGladwin@som.umaryland.edu; pcorti@som.umaryland.edu

enzyme, which is likely essential for maintenance of reduced Cygb in cells and catalytic electron transfer reactions[3]. Aged Cygb knockout mice have been reported to develop organ abnormalities including cardiac hypertrophy, kidney cysts, liver fibrosis and lymphoma[14], and decreased mean arterial blood pressure related to reduced NO metabolism[11]. While substantial research is underway to determine the physiological role of Cygb, questions remain regarding its precise function and its role in development has not been studied. To further probe for a physiological function of Cygb and a possible role in development we explored loss of function using CRISPR/Cas9 directed mutagenesis in the zebrafish model. Our results indicate that Cygb is essential for NO signaling and cilia function during development, required for correct left-right patterning.

## Results

### Cytoglobin is required for proper organ laterality determination

Since its discovery[15–17], Cygb has been shown to be ubiquitously expressed in nearly all human tissues but the mechanisms underlying Cygb function are unknown. Zebrafish carry two cytoglobin genes (*cygb1* and *cygb2*) but Cygb2 has higher identity/homology (63%/75%) to the mammalian cytoglobin[6] and similar heme hexacoordination as opposed to the pentacoordinate Cygb1[6]. *cygb2* mRNA is undetectable at initial embryonic developmental stages in the oocytes and at the 2–4 cell stage, but it is expressed during gastrulation, and further increases during somitogenesis, suggesting a role in development (Fig. S1a). To determine the physiological function of Cygb2, we

generated two *cygb2* mutant alleles *cygb2^801a* and *cygb2^801b* using CRISPR/Cas9. Two guide RNAs were designed in the first exon and successfully targeted a 4- and 1-base pair frame-shift mutation within the A helix of the globin fold that were predicted to form truncated proteins (Fig. 1a). Western blot analysis confirms reduced Cygb2 expression in mutants (Fig. S1b). Both *cygb2^801a and cygb2^801b* mutants developed organ laterality abnormalities with a significant decrease in embryos with normal, left-sided heart (87.3% ± 6.8 in wt, decreased to 58.7% ± 12.8 in *cygb2^801a* and 64.5% ± 8.9 in *cygb2^801b*) and increased in the opposite right-sided heart or straight looping in a cardiac bilateral symmetry (Figs. 1b–d, S1c; Supplementary movies: S1 and S2). Low levels of laterality defects (~10%) intrinsically develop in wt embryos and we observed this in our studies as well[18,19]. Expression of *southpaw* (*spaw*) mRNA, a left-side expressed nodal related gene, was present in a bilateral symmetry and right-sided pattern more commonly in *cygb2^801a* than wt siblings. The heart (labeled with *lefty2* and *mly7*), liver, and pancreas (both labeled with *foxa3*) had a greater frequency of midline and right-sided orientation compared to wt embryos (Fig. 1e, f). Human patients with primary ciliary dyskinesia (PCD) in some cases present with abnormal organ laterality[20]. The absence of functional cilia in PCD-affected subjects results in a random distribution of organs to left and right (50:50), similar to what we see in our *cygb2* mutant embryos. *cygb2* mutants also developed a shortened body axis that persists throughout adulthood (Fig. S1d, e). A significant curvature of the body axes was noticeable, (Fig. S1f) consistent with a scoliotic phenotype possibly induced by cilia

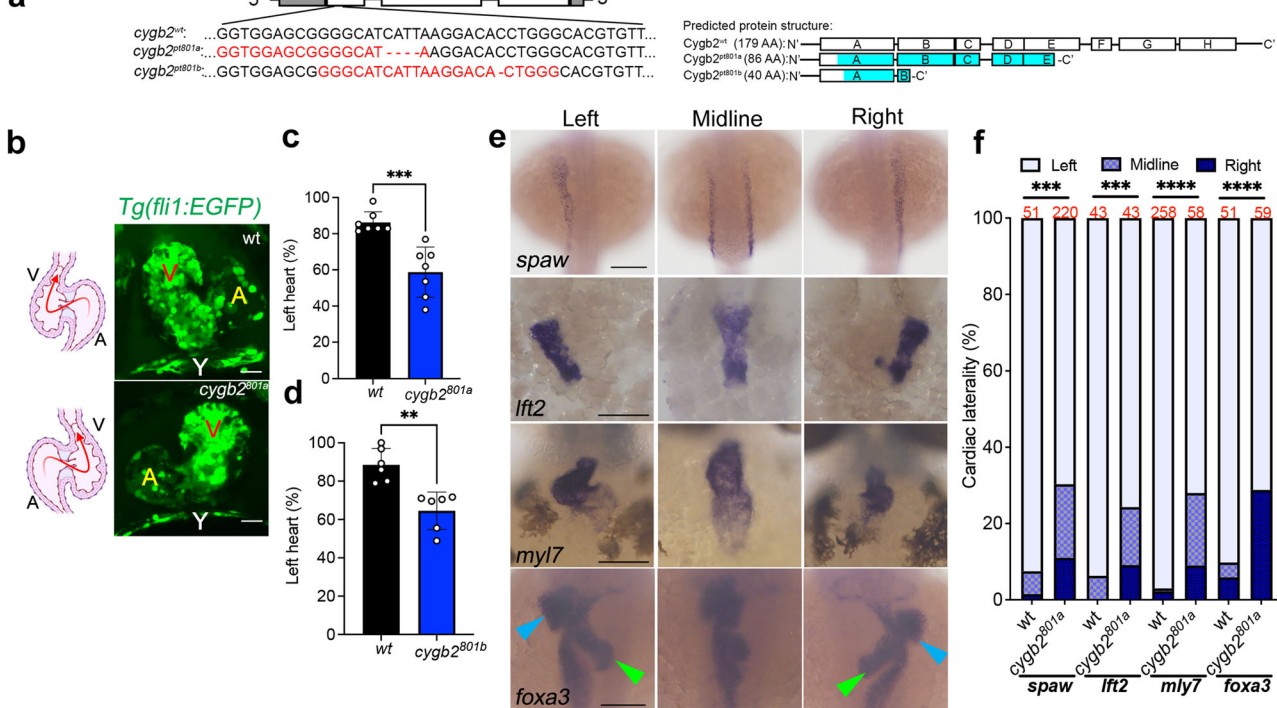

**Fig. 1 | *cygb2* mutant phenotype presents organ laterality defects. a** CRISPR/Cas9 mediated genome editing of *cygb2*. Two different gRNA were targeted to exon 1 (denoted by red text) and resulted in 4 bp and 1 bp frame shift mutations (beginning in the blue shaded region of the predicted protein structure) named *cygb2^801a* and *cygb2^801b*, respectively. The eight globin protein helices (labeled A-H) are represented by boxes, with out-of-frame amino acids shaded blue. **b** Whole mount 3D confocal projections (right) of wt and *cygb2^pt801a* *tg(fli1eGFP^y1)* hearts at 4 days post fertilization (dpf) with schematic (left) representing the heart morphology and direction of blood flow. V – ventricle, A – atrium, Y – yolk. Scale bar = 20 μm. **c, d** Quantification of the percentage of embryos with a left-sided heart loop in *cygb2^801a* and *cygb2^801b*. Means are ± SD (*n* = 6–7, each n representing an

independent experiment consisting of 50 embryos). Student's *t* test, two-tailed, **\**P < 0.01, ***P < 0.001. **e** Representative in situ hybridization images of the *cygb2^801a* laterality phenotype. *southpaw* (*spaw*), 16 somites, dorsal view; *lefty2* (*lft2*), 22 h post fertilization (hpf), dorsal view; *myosin light chain 7* (*myl7*), 96 hpf, ventral view; and *foxa3*, 2 dpf, dorsal view. Green arrow heads indicate the liver and blue arrow heads point to the pancreas. Scale bars = 100 μm. **f** Quantification of the percentage of embryos with right, straight/bilateral or left sided expression of *spaw*, *lft2*, *myl7* or *foxa3*. The total number of embryos analyzed is shown in red above the graph. The Chi-squared test was used to determine statistical significance. Source data are provided as source data file.

malfunction[21]. The probability of survival was normal in *cygb2*[801a] mutants (Fig. S1h).

Because of the high similarity between Cygb2 and Cygb1, we reasoned that the knockout of *cygb1* could also result in left-right abnormalities. Using CRISPR, we generated a *cygb1*[802a] maternal zygotic mutant line lacking Cygb1 protein (Fig. S1b, g) and analyzed survival rates and cardiac laterality: no significant defects in survival and cardiac left-right patterning were detectable (Fig. S1h, i). Additionally, Cygb1 protein expression is not altered in the *cygb2* mutants (Fig. S1b).

### Cygb2 co-localizes with cilia in the Kupffer's vesicle and regulates cilia structure

Vertebrate left-right development of organs is determined by ciliary beating within the embryonic node (ventral node in mammals and Kupffer's vesicle (KV) in zebrafish). The exact mechanism that initiates left-right asymmetry in the embryo is still a matter of debate. Different mechanosensing and chemosensing mechanisms have been proposed

to underlie the process, however, both theories agree that left-right gene expression is established as a consequence of a counterclockwise flow of fluid within the vesicle determined by cilia motility[22]. The observed laterality phenotype in *cygb2* mutants suggests a role for Cygb2 in KV function or KV cilia formation and/or motility. To localize Cygb2 in the embryo, we analyzed 8–10 somite embryos by immunofluorescence. Using two different anti-Cygb2 antibodies (Figs. S2a, S3a), we found colocalization of Cygb2 and acetylated tubulin, a marker of cilia, in the KV ciliated cells (Figs. 2a, S2a). *cygb2* transcript analysis by fluorescence whole mount in situ hybridization revealed *cygb2* expression in the KV ciliated cells colocalizing with *dand5* transcript, a marker of the KV (Fig. 2b).

Given the specific Cygb2 expression in cilia, we analyzed the length and number of cilia in the KV of *cygb2*[801a] and *cygb2*[801b] mutants. While cilia number remains the same (Fig. S3d), cilia length is markedly decreased in both *cygb2*[801a] and *cygb2*[801b] embryos (Figs. 2c, S2b). Of note, *cygb2* mutants have no defects in KV formation or size (Fig. S2c–e). This was confirmed by knockdown of Cygb2 with antisense

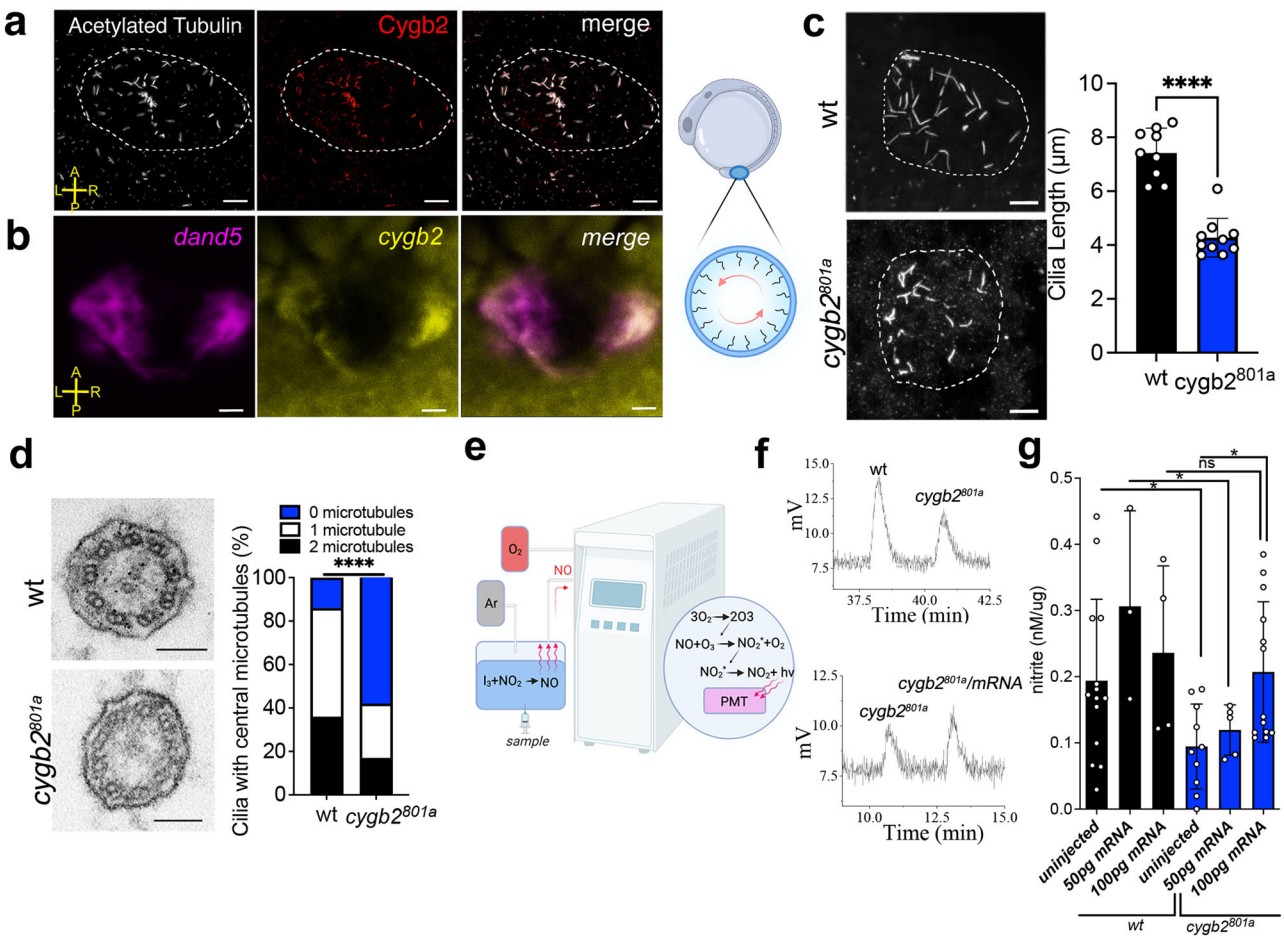

**Fig. 2 | Cygb2 is expressed in cilia and through NO production regulates cilia structure. a** Kupffer's vesicle (KV) at 8–10 somites stained with anti-acetylated tubulin (white) to visualize cilia and anti-Cygb2 (red) on whole mount embryos. The KV is outlined with a white dashed line. Scale bar = 10 μm. **b** Whole mount fluorescent in situ hybridization labeling *dand5* (magenta) and *cygb2* (yellow) transcripts using RNAscope. Image orientation: A- anterior, P- posterior, L- left, R- right. Scale bar = 10 μm. **a**, **b** show representative images of three experiments. On the right, rendering of a zebrafish embryo at 10 somites denoting the location of the KV. Panel generated with BioRender.com. **c** Immunostaining of KV cilia in wt and *cygb2*[801a] with anti-acetylated tubulin antibody. On the right, average cilia length. Means are ± SD (*n* = 9–10 embryos). Student's *t* test, two-tailed, ****P < 0.0001. Scale bar = 10 μm. **d** Transmission electron microscopy of KV cilia comparing wt and

*cygb2*[801a]. Scale bar = 100 nm. On the right, quantification of the number of central microtubules represented as the percentage of total number of cilia analyzed (*n* = 12–13 embryos). Chi-square test, ****P < 0.0001. **e** Schematic of the NO Analyzer (NOA). Panel generated with BioRender.com. **f** Chemiluminescence detection of NO production by triiodide assay using pools of 50 embryo (10 somites) lysates as samples comparing wt to *cygb2*[801a] (above) and uninjected *cygb2*[801a] to *cygb2* mRNA injected *cygb2*[801a] (below). **g** Quantification of nitrite levels normalized to protein amount. Samples represent pools of embryos (*n* = 3–14, each n is the lysate of 50 embryos) uninjected or injected with *cygb2* mRNA (50 or 100 pg). Black bars represent wt embryos and blue bars represent *cygb2*[801a] embryos. Means are ± SD. Student's *t* test, two-tailed, *P < 0.05. Source data are provided as source data file.

morpholino (MO), and 80% of *cygb2* injected embryos showed consistently shortened cilia (Fig. S3a–c). Similar to *cygb2^{801a}* and *cygb2^{801b}*, morphants also exhibit cardiac laterality defects (Fig. S3e, f).

To characterize the cilia morphology in the *cygb2* mutants, KV cilia ultrastructure was examined by transmission electron microscopy (TEM). Nine pairs of microtubule doublets as well as the inner and outer dynein arms were clearly visible in both wt and *cygb2^{801a}*. However, in the mutants there was a greater frequency of cilia present without a visible pair of central microtubules (Fig. 2d). This observation, along with the decreased length of cilia, suggests that mutations in Cygb2 alter either the formation or maintenance of cilia structure within the KV with possible consequences on cilia motility.

### Cytoglobin regulates NO levels in zebrafish embryos

As globins in general and cytoglobin in particular can regulate NO homeostasis[5,11,23,24], we used chemiluminescence assays optimized in our lab[25,26] to detect NO formation from embryos. The concentration of nitrite ($NO_2^-$) accumulating in embryo lysates is used as biomarker of NO formation, as NO oxidizes to the more stable $NO_2^-$. Injection of embryo lysates into a solution containing triiodide ($I_3^-$) quantitatively reduces $NO_2^-$ to NO, which is quantified by chemiluminescence. As the non-enzymatic, primary decomposition product of NO in tissues is $NO_2^-$, the $NO_2^-$ present in the samples linearly correlates with the NO that was produced in the tissue before harvest. We found that $NO/NO_2^-$ is lower in *cygb2^{801a}* compared to wt, affirming a possible role of Cygb2 in $NO/NO_2^-$ generation. Injections of *cygb2-mRNA* in *cygb2^{801a}* mutants restored $NO/NO_2^-$ to normal levels at early gastrula stage (Fig. 2e–g). While *cygb2-mRNA* did not rescue the laterality phenotype observed at 2 dpf, we conclude that the depletion strategies used in this study are specific for Cygb2 due to the reproducibility of the phenotype observed in two mutant alleles and in the morpholino knockdown model. Additionally, outcrossing the mutants over several generations decreases the possibility that Cas9-mediated off target effects underlie the observed phenotype. These data suggest that Cygb2 positively regulates NO formation.

### nos2b and gucy1a knock-down recapitulates cygb2 mutant phenotype

In aerobic conditions NO is mainly produced by the NO synthases (NOSs)[27] and typically signals by binding its canonical receptor soluble guanylate cyclase (sGC), which converts GTP to cGMP to regulate downstream signaling pathways[28] (Fig. 3a). In mammals, all NOS isoforms are found in motile cilia where they have been shown to regulate cilia motility[29]. While physiological levels of NO have been shown to facilitate cilia function, very low or high NO levels have been shown to impair cilia function[30], suggesting that NO levels must be finely regulated in order to ensure normal ciliary motility. Additionally, ciliopathies have been associated with low NO in airways and this is one of the hallmarks used for screening for PCD[31,32]. In zebrafish, *nos2b* is detectable in somitogenesis, with significantly lower expression observed in *cygb2^{801a}* mutants (Fig. 3b). Remarkably, antibody staining revealed co-localization of Nos2b with Cygb2 in KV ciliated cells (Fig. 3c). Fluorescent whole mount in situ hybridization was used to confirm that indeed, *nos2b* and *cygb2* transcripts are present in the same KV cells marked by *dand5* expression (Fig. 3d).

To determine the involvement of Nos2b with Cygb2 in laterality determination, we performed morpholino (MO) knockdown experiments and analyzed embryonic phenotypes. *nos2b*-MO injected embryos fully recapitulated the *cygb2* mutant phenotype, showing a decrease in cilia length and a significant decrease in embryos with left-sided heart ($88.9 \pm 5.2$ % in control embryos and $57.2 \pm 9.4$ % in *nos2b*-MO) with an increase in embryos displaying a right-sided heart or straight heart loop (Figs. 3e–g, S4a). The *nos2b*-MO phenotype was validated in our studies by simultaneous injections of a *nos2b-mRNA* construct with *nos2b-MO* in wt embryos where partial rescue of the cardiac laterality phenotype was observed (Fig. S4b). Our results suggest that Nos2b knock-down phenocopies the Cygb2 mutant, both appearing necessary for NO production in cilia.

To establish whether the NO-sGC-cGMP canonical signaling supports ciliary function we knocked down sGC using two MOs (*gucy1a*-MOs) and evaluated embryonic phenotypes. Both ATG and the previously validated splice blocking (SB)[33] MOs produced a shortened cilia length and a cardiac laterality phenotype with $56.9 \pm 5.1$% injected embryos showing left-sided heart compared to $88 \pm 5.4$% of control embryos (Figs. 3h–k, S4c–e). Consistent with a role for cytoglobin in increasing NO-dependent sGC activation, we observed lower cGMP concentrations in *cygb2^{801a}* mutants. cGMP levels could be recovered with the NO donor diethylenetriamine/NO (DETA/NO), indicating that the sGC enzyme itself remained functional (Fig. 3l). Altogether, our data imply that Cygb2 is a regulator of the Nos2b-NO-sGC-cGMP signaling axis, necessary for normal ciliary function and left-right patterning.

Another proposed function for Cygb is protecting cells from reactive oxygen species (ROS) damage, via a superoxide dismutase (SOD) activity[12]. The reaction of superoxide with NO is one of the fastest known physiological reactions and results in the formation of the highly reactive anion peroxynitrite thus limiting NO bioavailability. We observed a lack of SOD activity of recombinant Cygb2 (Fig. S5a, b), suggesting that Cygb2-dependent superoxide scavenging is not responsible for the mechanisms of cytoglobin-dependent NO signaling. Also supporting this observation, SOD and SOD-mimetic treatments failed to rescue cardiac laterality in *cygb2* mutant zebrafish (Fig. S5c, d). Additional in vitro experiments assessing the SOD activity of zebrafish Cygb1 and human CYGB failed to detect substantial SOD activity (Fig. S5e, f). Cygb1 showed only a weak SOD activity, albeit orders of magnitude lower than that observed for SOD (Fig. S5e, f).

### NO donor, sGC stimulator and nos2b-mRNA can rescue the mutant KV fluid flow and laterality phenotype in cygb2^{801a}

Motile cilia in the KV generates a leftward flow of fluid that is necessary for establishing the proper left-right asymmetry observed in vertebrates[34–36]. Based on the observation that *cygb2* mutants and morphants have shorter cilia, we hypothesized that laterality defects observed in the *cygb2* mutants were a result of altered fluid flow in the KV. To verify our hypothesis, fluorescent beads were injected into the lumen and we tracked flow rates over time. In wt embryos, beads moved in the characteristic counterclockwise direction. However, directionality of flow was disrupted, and bead velocity significantly decreased in a majority of *cygb2* mutants (Fig. 4a, b, and Supplementary movies S3–6).

To evaluate whether the effects of Cygb2 expression on ciliary function relate to NO, we studied fluid flow following treatment with the NO scavenger cPTIO or the NO donor DETA/NO. Wt embryos exposed to cPTIO showed a partial phenocopy of the mutant with decreased bead velocity, whereas the *cygb2^{801a}* exposed to DETA/NO displayed a partial rescue with increased bead velocity (Fig. 4a, b, and supplementary movies S3–6). DETA/NO treatment completely rescued the cardiac laterality defects in mutants at a concentration of 250 µM with no significant changes in cilia length (Figs. 4c, S6a, b). This effect is dose and stage specific when embryos are treated at the beginning of gastrulation (Fig. 4d, e). The precursors of the KV become detectable at this stage and migrate ahead of the dorsal blastoderm during gastrulation. These cells contribute to both the tailbud and the KV during somitogenesis[37]. Only treatments covering this time window (i.e. gastrulation/early somitogenesis) were effective and completely rescued the *cygb2* mutant laterality defects.

To further test the hypothesis that the downstream target of Cygb2-dependent NO signaling is indeed sGC, we treated embryos with the specific small molecule sGC activator BAY 582667 (cinaciguat), able to activate sGC, regardless of the presence of its ligand

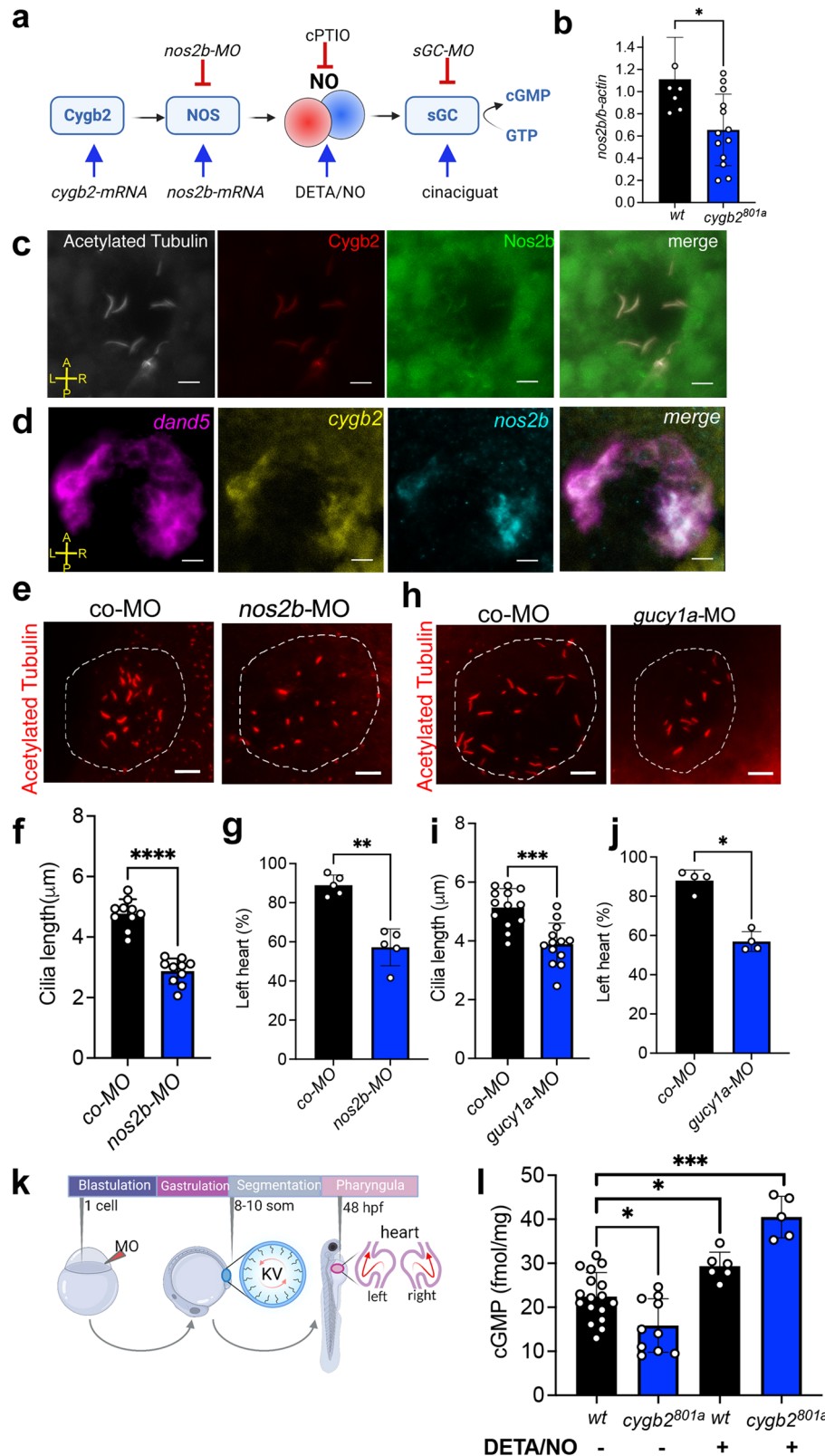

NO. The percentage of left-sided heart embryos significantly increased after administration of cinaciguat to *cygb2* mutants compared to controls (Figs. 4f, S6c). Lastly, to correlate the role of Cygb2 upstream of Nos2b we injected a *nos2b-mRNA* construct in *cygb2*[801a] mutants and rescued the cardiac laterality phenotype in 70.2% ± 7.0 injected embryos compared to 55.1% ± 11.0 of uninjected (Figs. 4g, S6d), comparable to the percentages obtained by injecting *nos2b-*

*mRNA* construct and *nos2b-MO* simultaneously into wt embryos (Fig. S4b).

### Cygb expression and requirement for NO formation and cilia function is conserved in mouse airway epithelium

To determine functional conservation of cytoglobin in mammals, we localized cytoglobin protein by western blot of human trachea scrapes

**Fig. 3 | Perturbations of the NO signaling pathway phenocopy *cygb2* mutants.**
**a** Schematic of Cygb2-NO-sGC signaling outlining experimental approaches to activate and inhibit the pathway. Panel generated with BioRender.com. **b** mRNA expression of *nos2b* relative to *β-actin* at 8–10 somites normalized to wt expression quantified by qRT-PCR. Means are ± SD (*n* = 6–12, each n is the lysate of 50 embryos). **c** Immunostaining of acetylated tubulin (white), Cygb2 (red) and Nos2b (green) on whole mount embryos. **d** Whole mount fluorescence in situ hybridization labeling *dand5* (pink), *cygb2* (yellow) and *nos2b* (cyan) transcripts using RNAscope. **c, d** show representative images of three experiments.
**e, h** Immunostaining of acetylated tubulin in the KV of scrambled control and *nos2b* or *gucy1a* morphants. The KV is outlined with a white dashed line. **f, i** Average cilia length, means are ± SD (*n* = 10–13 embryos). Student's *t* test, two-tailed,

****P < 0.0001, ***P < 0.001. **g, j** Percentage of embryos with left-sided hearts in control morphants (co-MO) versus *nos2b* (ATG MO) and *gucy1a* (ATG MO) morphants, means are ± SD (*n* = 4–5, each n representing an independent experiment consisting of 50 embryos). Student's *t* test, two-tailed, **P < 0.01, *P < 0.05.
**k** Experimental schematic demonstrating the timing of morpholino (MO) injection, KV formation and left/right patterning phenotype. Panel generated with BioRender.com. **l** cGMP levels were measured by ELISA comparing wt to *cygb2*[pt801a] with and without 250 μM DETA/NO treatment. Means are ± SD (*n* = 5–17 independent replicates with each sample representing 50 pooled embryos at 8–10 somites). Student's *t* test, two-tailed, *P < 0.05, ***P < 0.001. Image orientation: A- anterior, P- posterior, L- left, R- right. Scale bar = 10 μm. Source data are provided as source data file.

obtained from two healthy human subjects. The band corresponding to CYGB (~21 kDa) was detectable in both scrapes from human trachea (Fig. S7a). Similarly, analysis of human lungs by single cell RNA-Seq recently made available in the Human Protein Atlas[38] reveals the presence of CYGB in airway ciliated epithelial cells. We also confirmed that Cygb is expressed in mouse airway ciliated epithelial cells by immunostaining and western blot (Figs. 5a, S7b, c). Like zebrafish Cygb2 mutants, global knockout of Cygb in mouse presented shorter airway cilia (Fig. 5b). Additionally, by chemiluminescence analysis of mouse tracheas, we found decreased NO/NO$_2^-$ levels in knockouts compared to wt mice (Fig. 5c). Murine ciliated cells express *iNOS* and *eNOS* isoforms with no significant differences in expression levels in knockout whole trachea lysate (Fig. S7d). However, the fluorescence intensity of *eNOS* and *iNOS* transcripts specifically measured in the ciliated epithelium analyzed by fluorescence in situ hybridization shows a significant decrease in *eNOS* but not *iNOS* (Fig. S7e, f). Of note, Nos2b alignment of the deduced amino acid sequences shares high sequence identity with both mammalian iNOS (58% identity) and eNOS (51% identity).

These data suggest a conserved function for cytoglobin in the regulation of NO synthesis and cilia function in the airway. Increased NO production in wt compared to knockout suggests possible positive interactions between NOS and cytoglobin in producing NO. In order to verify electron transfer between Cygb and NOS in vitro, we measured the reduction of human ferric CYGB by mammalian iNOS or zebrafish Nos2b. The incubation of CYGB with NADPH alone can produce spectral changes consistent with the formation of the ferrous CYGB-Fe$^{2+}$-O$_2$ complex, indicating the reduction of ferric CYGB to its ferrous state (Fig. 5d, left). However, in the presence of mammalian iNOS or zebrafish Nos2b the reduction is much faster, indicating a specific NOS/CYGB interaction (Fig. 5d, e). To determine whether CYGB could directly impact NOS activity to produce NO, we measured the formation of L-citrulline by NOS in the presence or absence of CYGB. L-citrulline is a stochiometric byproduct of NO formation by NOS enzymes. In aerobic conditions, the rates of NO synthesis by iNOS did not increase in the presence of CYGB. However, when the oxygen tension was reduced to 3.2% (approx. 40 μM O$_2$) (Fig. 5f) we observed significantly higher L-citrulline levels in the presence of CYGB suggesting enhanced NO production by iNOS in the presence of oxygen-bound CYGB (Fig. 5f).

## Discussion

Despite significant research on Cytoglobin since its discovery, questions remain about its physiological functions. We find that genetic deletion of *cygb2* in zebrafish causes cardiac and gastro-intestinal tract laterality defects, which in humans is associated with airway motile cilia defects and low nasal nitric oxide (NO) levels. This study reveals that Cygb2 co-localizes with cilia and the NO synthase (NOS) Nos2b in the KV, and the structure and function of cilia were disrupted in *cygb2* mutants, abolishing fluid flow within the KV. NO and nitrite levels are reduced in the Cygb2 knockout and abnormal ciliary function and organ laterality is phenocopied by depletion of *nos2b* and *gucy1a*, one

of the NOS isoforms and the canonical NO receptor soluble guanylate cyclase (sGC) homologs in fish, respectively. Cilia function is rescued by exposing *cygb2* mutant embryos to a NO donor, a sGC stimulator and with over-expression of *nos2b*. These studies in aggregate support the existence of a cytoglobin-NOS-NO-sGC-cGMP signaling axis that regulates cilia structure and function. Consistent with a conserved role in regulating cilia function, *Cygb* knockout also impaired mouse airway epithelial cilia structure and reduced NO production. Thus cytoglobin appears essential for normal development stage-specific NO signaling and ciliogenesis.

Primary ciliary dyskinesia (PCD) encompasses a growing class of disorders caused by abnormal ciliary axonemal structure and function[20]. Unlike many mendelian genetic disorders, PCD is not caused by mutations in a single gene or locus, but rather autosomal, recessive mutation(s) in one of many genes that can lead to a similar phenotype characterized by chronic infections of the respiratory tract, male infertility and organ laterality defects[39,40]. To date, more than 50 PCD-associated mutations have been identified, accounting for only 70% of PCD patients. Cytoglobin was recently identified in a recessive forward genetic screen in fetal mice associated with hypoplastic left heart syndrome[41]. Genetic testing is used for diagnosis along with the measurement of nasal exhaled NO levels, with low to undetectable levels of NO being one of the hallmarks of PCD[32]. Despite extensive studies, a mechanism accounting for low exhaled NO levels in patients with PCD has never been established, nor is it clear whether the reduced production of NO is a cause or a consequence of ciliary dysfunction. Our data suggests that cytoglobin is an upstream positive regulator of cilia function through activation of NOS-NO-sGC-cGMP. As such it is possible that a primary mutation in cilia genes may be associated with compensatory upregulation of the CYGB-NOS-NO-sGC pathway.

In airway epithelial cells, all the key elements of the NO signaling pathway have been identified within a ciliary "metabolon" located at the apical surface of the cell at the base of cilia[42]. This includes the three NO synthase isoforms and the NO receptor soluble guanylate cyclase (sGC), which binds NO with high affinity and when activated catalyzes the conversion of GTP to cGMP. A number of studies have examined cGMP-mediated stimulation of cilia beat frequency in a NO- and sGC-dependent manner [reviewed in[43]]. Although these data suggest a role for NOS enzymes in cilia function, the relationship between NOS function and cilia motility is still unclear, with both high and low levels correlating with ciliary dysfunction. Our results provide strong support for the requirement of NOS-sGC-cGMP signaling in the assembly and function of working cilia, and suggests that low exhaled NO in PCD is more than a biomarker of disease.

Androglobin is a newly discovered globin that appears to be related to ciliogenesis[44], however no globin has been known to regulate ciliary function. Although the redox sensor properties of cytoglobin and electron transfer capacity suggest an important role in redox signaling[12], in our studies Cygb2 showed no SOD activity and was associated with lower NO and nitrite levels, suggesting alternative signaling properties in cilia. Cytoglobin can directly regulate NO levels

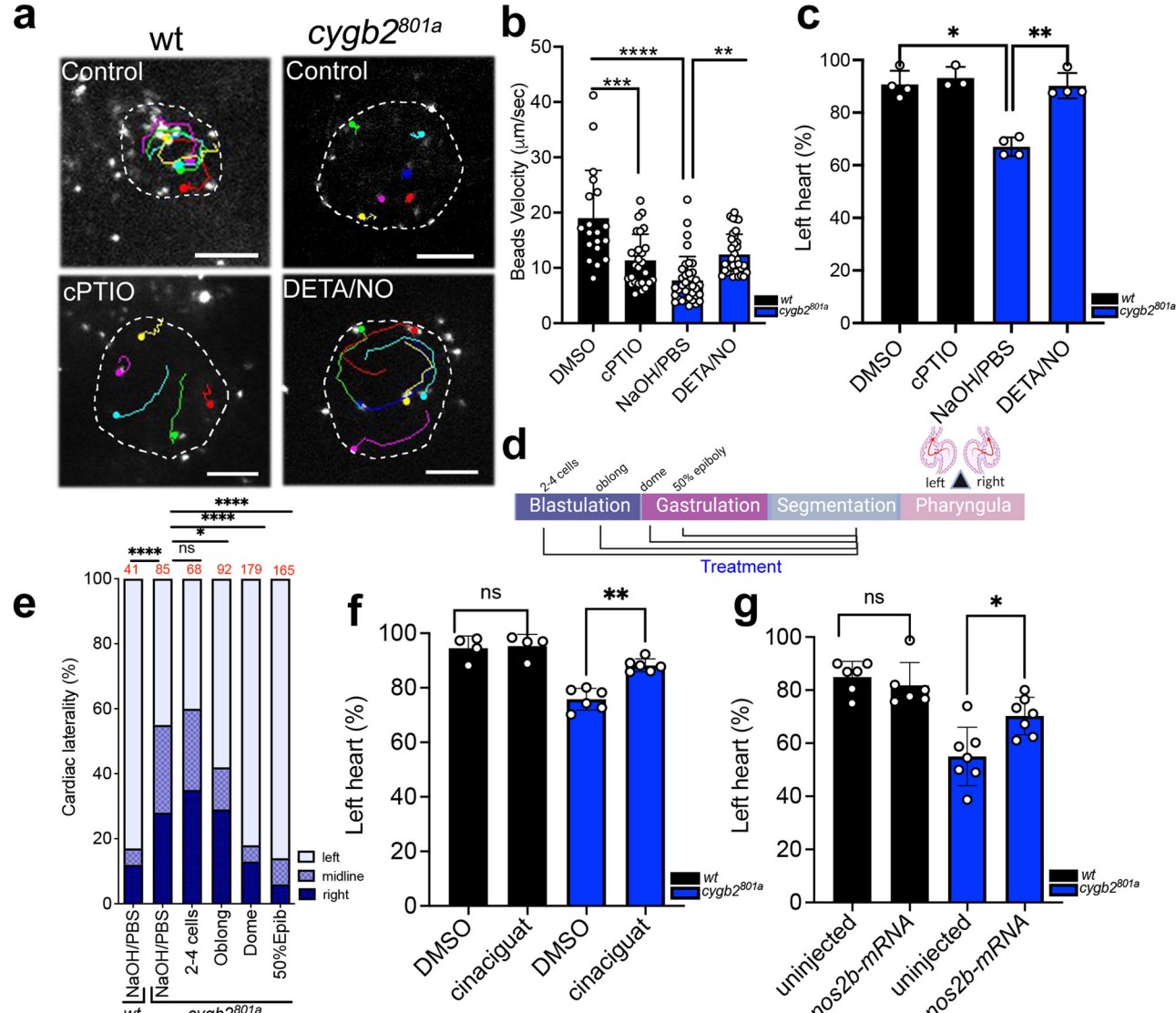

**Fig. 4 | Cygb2 regulates Nos2b-NO-sGC signaling in KV cilia function and cardiac laterality determination. a** Beads were injected into the KV between 8–10 somites and velocity was measured over time. Representative bead tracks of the KV comparing wt, *cygb2[801a]* embryos, wt treated with 500 μM cPTIO and DMSO as control and *cygb2[801a]* treated with 250 μM DETA/NO and NaOH/PBS as control (*n* = 4 embryos per treatment). The dot indicates the end of the bead track after ten seconds of tracking. Scale bar = 20 μm. Boundaries of the KV are demarcated by a white dashed line. **b** Quantification of bead velocities (*n* = 20–35 beads over 4 embryos). Means are ± SD. One-way Anova, ***P* = 0.003; *****P* < 0.0001. **c** Analysis of cardiac laterality defects represented as a percentage of embryos with the correct heart looping (on the left) analyzed at 2 dpf. Wt and *cygb[801a]* treated with 250 μM DETA/NO or 500 μM cPTIO at the end of the blastula period. Means are ± SD, (*n* = 4). Student's *t* test, two-tailed, ns = not significant, **P* < 0.05, ***P* < 0.01.

**d** Experimental schematic of the developmental time course drug treatment with 250 μM DETA/NO. Panel generated with BioRender.com. **e** Percentage of embryos with right, midline or left heart orientation following treatment with DETA/NO beginning at different developmental stages. The total number of embryos analyzed is shown in red above the graph. The Chi-squared test was used to determine statistical significance. **f** Quantification of the percentage of embryos with left hearts comparing wt to *cygb2[pt801a]* with and without treatment of 80 μM cinaciguat. Means are ± SD, (*n* = 4–6). Student's *t* test, two-tailed, ns = not significant, ***P* < 0.01. **g** Quantification of the percentage of embryos with left heart comparing wt to *cygb2[pt801a]* injected with *nos2b* mRNA compared to uninjected controls. Means are ± SD, (*n* = 6–7). Student's *t* test, two-tailed, **P* < 0.05, ns = not significant. In (**c**, **f**, **g**): each sample represent an independent experiment consisting of 50 embryos. Source data are provided as source data file.

in two opposing ways: (i) NO generation: reduced CYGB ($Fe^{2+}$) can produce NO by nitrite ($NO_2^-$) reduction, and (ii) NO depletion: $O_2$-bound Cygb reacts at diffusion-limited rates with NO (NO dioxygenation) producing nitrate ($NO_3^-$)[10,11] as final species. Our data supports an (iii) *indirect mechanism* where Cygb either interacts with, and stimulates NOS-dependent NO formation from L-arginine oxidation by improving oxygen delivery to NOS, or uses NOS as a reductase to support NO generation from nitrite. In the current studies we find that Cygb2 and Nos2b need to be simultaneously present for proper NO production and cilia function. NO levels are tightly regulated in cilia, and Cygb could work in a concerted manner with NOS by the

mechanisms described above, to finely tune NO levels to the optimal values for cilia function. Importantly, there are no examples of a globin positively regulating NO synthases to date.

In vertebrates, some internal organs are positioned asymmetrically across the left-right axis. Events determining left-right asymmetry during embryonic development are dependent on motile cilia function[20,34,45] and are regulated by mechanisms evolutionarily conserved among vertebrates[35,36]. Mechanistically, cilia beat within the left-right organizer generates a circular flow of fluid within the vesicle. This circular flow drives the asymmetric gene expression that instructs the symmetry of developing organs. Thus, ciliopathies

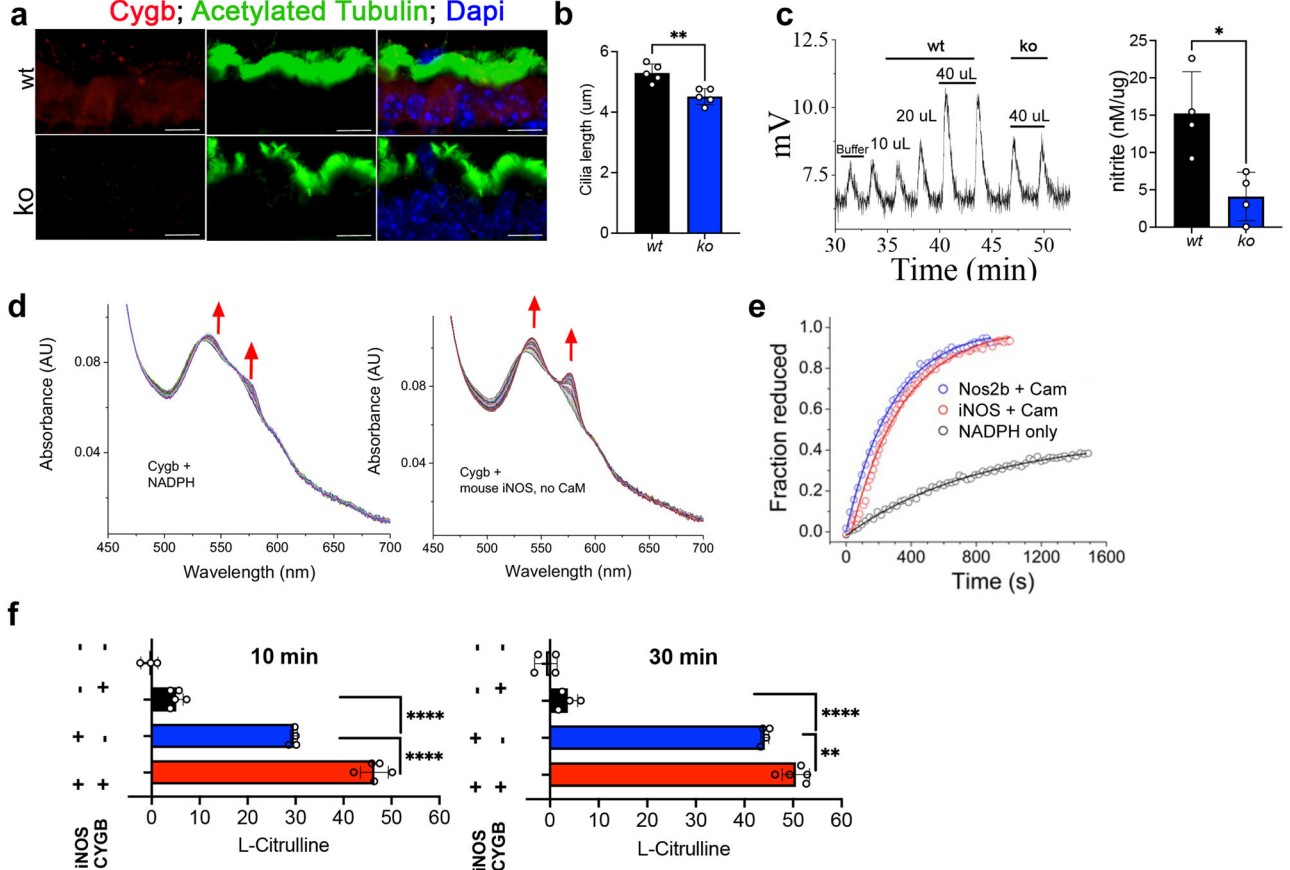

**Fig. 5 | Cygb is required for NO formation through molecular interactions with iNOS. a** Immunostaining of Cygb in mouse airway epithelium visualized by immunofluorescence. Scale bar = 10 μm. **b** Quantification of cilia length from wt and *Cygb* mutant (ko) mouse (*n* = 4). Means are ± SD. Student's *t* test, two-tailed, **P** < 0.01. **c** Chemiluminescence detection of NO production by triiodide assays using mouse trachea lysates as samples comparing wt to ko with injection volume indicated. On the right, quantification of nitrite concentration normalized by protein amount in each tissue (*n* = 4). Means are ± SD. Student's *t* test, two-tailed, *P* < 0.05. **d** Reaction of 8 μM ferric CYGB in the presence of 250 μM NADPH.

Increased absorbance at around 543 and 575 nm is consistent with the reduction of CYGB and formation of the CYGB Fe$^{2+}$-O$_2$ complex. On the left, the reaction with CYGB and NADPH only. On the right, the same reaction in presence of 12.5 nM mouse iNOS. **e** Traces indicating the reduction of ferric CYGB in the presence of different reducing systems: zebrafish Nos2b, mouse iNOS or NADPH alone. **f** L-Citrulline assay indicating NO production by iNOS in presence or absence of CYGB (*n* = 3–5). Means are ± SD. Student's *t* test, two-tailed, **P** < 0.01, ****P** < 0.0001. Source data are provided as source data file.

in humans are associated with developmental laterality defects[46,47]. In humans with PCD, the levels of exhaled NO are almost unmeasurable, and in chick development, the endogenous addition of pharmacological NO induces laterality defects[48]. Our findings indicate that NO signaling directly regulates ciliary function and that a CYGB-NOS-NO-sGC-cGMP signaling axis controls ciliary function in zebrafish development. Gain and loss of function studies suggest that cytoglobin expression is essential for normal ciliary function, with cytoglobin positively regulating NO signaling and the development of organ laterality.

In conclusion, the advent of genome sequencing resulted in the discovery of a host of previously unidentified globin proteins expressed outside of red blood cells and skeletal muscle, changing the paradigm through which we understand this well-studied protein family. While heme globins such as cytoglobin are highly conserved in vertebrates, their physiological functions have been unknown, largely related to inconsistent and mild to absent observed phenotypes in mice. Here we show the presence of a globin associated with cilia that is required for appropriate NO signaling in development of organ laterality through regulation of cilia structure and function within the vertebrate left-right organizer. Overall, these findings identify non-canonical functions for members of the globin superfamily and lay the foundation for mechanistic studies of how cytoglobin, NOS, NO and

sGC regulate cilia function, and identify the Cygb-NOS-NO-sGC-cGMP signaling pathway as target for therapeutic interventions in ciliopathies.

## Methods
### Animal care
Adult zebrafish (*Danio rerio*) were maintained according to standard protocols and according to protocols approved by the University of Pittsburgh Institutional Animal Care and Use Committee (Protocol # 21059400) and University of Maryland Institutional Animal Care and Use Committee (Protocol # 1022015), conforming to NIH guidelines. AB was used as the wild type genetic background. Embryos were kept in E3 embryo medium supplemented with 0.003% phenylthiourea (PTU) at 24 h post fertilization (hpf) to prevent melanin synthesis for embryos that were used for imaging. Embryos were grown at 25 °C to allow the analysis in vivo of the developmental stages of interest. The transgenic lines used in this study have previously been described: *(Tg(fli1:EGFP)$^{y1}$)*[49]; *Tg (sox17:GFP)$^{s870}$*[50] and *Tg (myl7:DsRed)$^{s879}$*[51].

All mouse (*Mus musculus*) experiments were approved by the University of Pittsburgh Institutional Animal Care and Use Committee (Protocol # 21099953) and University of Maryland Institutional Animal Care and Use Committee (Protocol # 1022015). Mice were housed at 21–23 °C with 30–70% humidity and a 12 h dark/light cycle in a

pathogen-free facility. Mouse strains used in this study were wild-type and Cygb knockout in the C57BL/6 background. The mice were all 20 month old females. The Cygb knockout mouse strain was developed by Dr. Anders Hay-Schmidt (University of Copenhagen, Denmark). A constitutive knockout allele was developed from a Cygb conditional knockout mouse via deletion of exon 2. Heterozygous constitutive knockout F1 progeny were obtained, backcrossed with C57BL/6 J mice for 7 generations before being used for experiments[52].

### cygb2 mutant generation, genotyping and morpholino injection

A genomic deletion was produced in the first exon of zebrafish cygb2 and cygb1 using CRISPR-Cas9 genome editing. Target sites were selected using online software tools[53] and are shown in Fig. 1 a and Supplementary Fig. S1G. Sequences of gRNA primers and of primers for screening mutations are listed in Table S1. For Cygb2 knockout we recovered two F1 adult fish carrying a 4 bp and a 1 bp deletion with frame-shift mutations and established two stable lines, cygb[801a] and cygb2[801b] respectively. For Cygb1 knockout we recovered one F1 adult fish carrying a 8 bp deletion with frame-shift mutations and established a stable line (cygb1[802]). The F1 adult fish carrying the recovered alleles and the ones shown to be wt were outcrossed with uninjected fish, the progeny were screened for the mutation of interest and the mutant fish were bread to F3/F4 generation before using them for experiments. The wt controls are siblings selected at the time of the recovered mutation. The gRNA was made according to published protocols[54,55], using the Megashort SP6 Transcription kit (Thermo-Fisher, AM1330) followed by RNA purification with the Megaclear Kit (ThermoFisher AM1908). Cas9 mRNA was synthesized from pCS2-nCas9n using the mMessage mMachine SP6 Transcription kit (ThermoFisher, AM1340). A mixture of gRNA (300 pg) and Cas9 mRNA (600 pg) was coinjected at the 1 cell stage. Embryos were raised and screened for restriction site disruption (pt801a – MseI, pt801b – PspGI) using primers specific to exon 1 of cygb2 (Table S1). Mutagenesis was verified by sequencing of genomic DNA using homozygous mutant embryos.

To knockdown the translation of Cygb2, sGCa, and Nos2b, we used morpholino (MO) knock-down strategies. Morpholinos were designed by Gene Tools and the injected dose is specified as follow: cygb2 AUG-MO (3 and 7 ng), gucy1a1 AUG-MO (5 ng), gucy1a1 splice blocking (SB) MO (1 and 2 ng), nosb2 AUG-MO (4 ng), standard control morpholino (5 ng). The nosb2 AUG-MO was validated by co-injections of nos2b mRNA showing partial rescue of the laterality phenotype (Fig. S4b). The gucy1a1-SB MO was previously validated[33,56] and its specificity was verified by transcript disruption in injected embryos (RNA was isolated from embryos at the 8–10 somite stage) (Fig. S4e) and validated by injections of an additional gucy1a1-ATG MO resulting in a similar laterality phenotype (Fig. S4c, d).

### cygb2 and nosb2b mRNA synthesis and overexpression

The cygb2 transcript was amplified from cDNA prepared from 3-day post fertilization embryos. Primers were designed to include a Cla1 restriction site and kozak sequence to the forward primer and an Xho1 site to the reverse primer. pCS2+ plasmid was digested with ClaI (NEB, R0197s) and Xho1 (NEB, R0146s) and ligated with the cygb2 PCR product. nos2b mRNA overexpression plasmid was constructed using GenScript cloning services. Briefly, the nos2b coding sequence (https://www.ncbi.nlm.nih.gov/nuccore/NM_001113501.1/) was cloned into pCS2+ plasmid using restriction cloning with Cla1 and Xba1 sites with the inclusion of the kozak sequence upstream of the start ATG (table S1). Clones were confirmed by sequencing. mRNA was generated using mMessage mMachine SP6 (Invitrogen, AM1340) on plasmids that had been linearized with Not1 (NEB, R0189s). Embryos were injected at the one cell stage with 50-100 pg cygb2 mRNA or 100 pg nos2b mRNA.

### RNA extraction and quantitative RT-PCR

RNA was isolated from pooled oocytes or embryos (50/sample) or mouse trachea. Tissues lysates were prepared using TRIzol Reagent (ThermoFisher). 1 ug of total isolated RNA was used in reverse transcription by SuperScript II Reverse Transcriptase (Thermofisher) and 50 ng of the resulting cDNA was used as template for TaqMan quantitative PCR as before[26]. The qPCR reaction used TaqMan Universal PCR Master Mix (Applied Biosystems) and TaqMan gene expression assays (Thermo Fisher): cygb2 (Dr03090731-m1), nos2b (Dr03431350_m1), iNOS (Mm00440502_m1), eNOS (Mm00440502_m1). All assays were done in triplicate and appropriate non-transcriptase and non-template control reactions were included. For relative quantification of the transcript, we used polr2d (assay Dr03095552-ml, Thermo Fisher) as a housekeeping gene for zebrafish samples and 18 s (assay Hs.PT.39a.22214856.g, IDT) for mouse samples. Results were collected with Quantstudio5 software and analyzed by the delta/delta $C_T$ method following triplicate averaging. Statistical comparisons between replicate pairs were determined by a paired, two-tailed Student's $t$ test.

### In situ hybridization

Whole mount in situ hybridization was performed as previously described[57] using digoxigenin-labeled riboprobes for cytoglobin 2 (cygb2), southpaw (spaw), lefty2 (lft2), myosin light chain 7 (myl7), forkhead box A3 (foxa3), and myogenic differentiation 1 (myod1). Transcripts were amplified from cDNA obtained from pooled embryos with a T7 polymerase site on the reverse primer (Table S1). The myod1 riboprobe was received as a gift from the Roman lab at the University of Pittsburgh. Images were acquired with an MVX-10 Macro View microscope and DP71 camera and compiled with Adobe Photoshop CS2 version 9.0.2 (Adobe Systems, San Jose, CA, USA). To determine laterality abnormalities all the stained embryos were analyzed and categorized in left cardiac loop, straight loop with cardiac bilateral symmetry or right cardiac loop group and percentages were calculated.

For fluorescence in situ hybridization we used RNAscope. Whole mount embryos at 8–10 somites and paraffin embedded mice trachea tissue slices were processed according to the Advance Cell Diagnostics protocol[58] and we used RNAscope Multiplex Fluorescent Reagent Kit V2 (Advance Cell Diagnostic 323100) with minor modifications detailed herein. 8–10 somite embryos were fixed for 1 h in 4% paraformaldehyde (PFA) at room temperature and dechorionated manually. Following washes in phosphate buffered saline with 0.1% tween (PBT), embryos were dehydrated in a methanol series diluted in 0.1% PBT and stored in 100% methanol at -20 °C for at least one night. Embryos were allowed to air-dry at RT for 30 min and subjected to the RNAscope-based signal amplification. Protease digestion of embryos using Pretreat 3 solution[59] was added and incubated at RT for 20 min followed by 0.01% PBT (0.01% Tween-20 in PBS, pH 7.4) washes. The hybridization of target probes was performed as per kit instructions. The following probes were used: Dr. nos2b-C1, Dr. dand5-C2, Dr. cygb2-C3, Mm Cygb-C3, Mm iNos-C1, and Mm eNos-C1. Embryos were washed in 0.2X SSCT (Fisher Scientific BP13251-1) and fixed for 10 min in 4% paraformaldehyde at room temperature. Embryos were incubated with different amplifier solutions for 30 min at 40 °C according to the protocol[58]. Horseradish peroxidase (HRP) signal was developed sequentially for 30 min for each probe in the dark with intermediate washes in 0.2X SSCT. We labeled HRP-C1 with Opal 520 diluted 1/750 in TSA buffer (provided in the kit), HRP-C2 with 1/750 Opal 570 and HRP-C3 with 1/750 Opal 620.

Mouse trachea tissue used for RNAscope was fixed with 10% Formalin for 24 h and processed by paraffin embedding procedure. Sections (4uM) were collected and backed for 1 h at 60 °C, then followed by xylene and serial ethanol hydration steps. The RNAscope procedure was then applied according to the kit instructions and the Advance Cell Diagnostics protocol. Confocal 3D projections images

were captured and maximum projections were processed using ImageJ.

## Immunostaining and western blot

Embryos used for KV visualization were fixed in 4% PFA for 1.5 h at RT. After dechorionation, embryos were permeabilized with 3%$H_2O_2$ in 100% methanol for 20 min at RT. Embryos were washed with 1X PBS-0.1%Triton-X 100, then incubated in blocking solution (0.1% Triton-X 100, 5% normal goat serum in PBS) for 1 h. To label Cygb2 we used two custom anti-Cygb2 antibodies (1:200 dilution, Thermo Scientific, Pierce Protein Biology): an anti-Cygb2 polyclonal antibody produced in guinea pig (used throughout the manuscript except where specified) and an anti-Cygb2$_{113-130}$ peptide antibody produced in rabbit (used in Fig. S2a) generated using recombinant Cygb2 that was previously obtained[2]. To label cilia we used and anti-acetylated tubulin (1:500, Sigma T7451) and to label Nos2b we used an anti-iNOS (1-200, NOVUS NB300, 605ss). The following secondary antibodies used were used diluted 1:1000: anti guinea pig AlexaFluor488 (ThermoFisher A-11073), anti-rabbit AlexaFluor 488 (ThermoFisher A-11034), anti-mouse AlexaFluor 488 (ThermoFisher A-11001), anti-rabbit Cy3 (Thermo-Fisher A-10520), anti-mouse Cy5 (ThermoFisher A-105254). Confocal 3D projections images were captured and maximum projections were processed using ImageJ. Cilia length was measured with at least 15 cilia per embryo measured and averaged, representing a single data point.

To stain mouse trachea, we used an anti-Cygb antibody (Sigma HPA017757 1:25 dilution, Santa Cruz 66855, 1:500 dilution) on paraffin slices and performed DAB staining or immunofluorescence for protein visualization. After deparaffinization sections (4um) were treated for antigen retrieval with Citrate buffer (10 mM sodium citrate buffer with 0.05% Tween-20, PH 6.0) at 90 °C, 3 times for 2 min each. Sections were incubated in blocking solution (0.1% Triton-X, 100, 5% normal goat serum, 3% BSA in PBS) for 1 h and incubated with anti-Cygb antibodies overnight at 4 °C. Secondary antibodies were used as 1/1000 diluted in blocking buffer and incubated at 4 °C overnight. For the DAB staining we proceeded following standard protocols and at the end of the procedure, in order to visualize the tissue, we added hematoxylin blue on the slides for 1 min prior to mounting the slides for imaging.

Western blotting procedure was followed according to standardized protocols using adult zebrafish brain lysates, mouse trachea lysates and human cells scraped from isolated tracheas as samples obtained by fresh homogenization in protein extraction buffer (RIPA buffer, Santa Cruz) supplemented with protease and phosphatase inhibitors. Aliquots of total lysates (150 µg) were loaded onto a precast mini-PROTEAN gel (Bio-Rad 456-8123). Transfer to a PVDF membrane was performed for 2 h on ice. After blocking in 5% skim milk-PBST for 1 h at RT, the PVDF membrane was probed with anti-Cygb2 polyclonal guinea pig antibody (Thermo Scientific, Pierce Protein Biology, 1:200 dilution), human anti-CYGB (Sigma HPA017757 1:200 dilution) and rabbit anti-beta-actin (A2066 Sigma, 1:200 dilution). Secondary HRP-conjugated antibody was used (1:1000 dilution, Abcam ab6789) and the relative levels of Cygb2 protein were normalized by beta-actin amounts. Data were captured using ChemiDoc MP Imaging system equipped with Image lab software (BioRad).

## Image acquisition and analysis

Live zebrafish embryos were imaged by confocal or bright-field microscopy at 8–10 somite stage for KV size and at 2–4 days post fertilization (dpf) for cardiac laterality determination. Embryos were removed from chorions and raised in 0.003% 1-phenyl-2-thiourea (PTU, Sigma P7629) starting after somitogenesis to prevent pigmentation. Live embryos were anesthetized in 0.1% tricaine, mounted in 1.5% low melt agarose and imaged by confocal microscopy. The fluorescent images were captured with ZEN software using an upright

Zeiss LSM 200 confocal microscope with a 20X water objective. Two-dimensional projections were generated from a Z-series. The movies and the bright field images were captured with Olympus LS software, using an Olympus MVX-ZB10 equipped with a DP74 color CMOS camera. Images were analyzed using ImageJ software.

Embryos for KV flow analysis were pressure injected with FITC-labeled fluorescent beads (0.5 µm) (Polysciences 18859-1), diluted 1:100 in sterile water. Intact embryos with detectable fluorescence in the KV after injection were mounted in glass bottom petri dishes (Mattek) in 1.5% low melt agarose and imaged on a spinning disc inverted confocal microscope (Leica DMI6000CS and Yokogawa CSUX-1) equipped with a Hamamatsu C9100-23B camera using a 25X water objective. Movies were made after collecting 2000 frames at 100 frames/s. Particle tracking was performed using Image J software as per the methods described[60]. Beads were tracked manually using the FIJI ImageJ plugin.

The analysis of cardiac symmetry and loop laterality orientation was conducted on live 2 dpf embryos. We determined percentages of left-bilateral-right sided heart embryos over total number of embryos collected. Biological replicates per genotype were combined. For treatment or injection experiments, we calculated percentages obtained after each experiment (total of 50-60 embryos per experimental condition) and averaged at least 4 different experiments. Unpaired, two tailed Student $T$ test and Chi-squared test were applied to detect statistical differences between conditions in GraphPad Prism version 9.

The spine curvature analysis was performed after treating the whole adult fish with 5% formalin, 5% Triton X-100 and 1% potassium hydroxide for 48 h at 42 °C, then transferred into a solution of 20% glycerol, 5% Triton X-100 and 1% potassium hydroxide and incubated at 42 °C for 60 h. Specimens were transferred to a solution of 0.70% Alizarin Red, 1% potassium hydroxide and incubated over night at room temperature prior to imaging.

## Drug exposure

Embryos were treated beginning at the dome stage until 24 hpf unless otherwise specified. Carboxy-PTIO potassium salt (cPTIO, Sigma – C221), N-Nitro-L-arginine methyl ester hydrochloride (L-NAME, Sigma – N5751), and BAY 582667 (Cinaciguat, Sigma – SML1532) were diluted in 100% DMSO and used at a final concentration of 0.1% DMSO. Diethylammonium (Z)-1-(N,N-diethylamino)diazen-1-ium-1,2-diolate (DETA/NO, Cayman Chemical Company – 82100) was resuspended in sodium hydroxide and diluted in phosphate buffered saline. SOD mimic (Fischer Scientific HY-13336) diluted in E3. All drug exposure doses are specified in the figure or figure legends.

## Nitric oxide analyzer

Since the primary non-enzymatic decomposition product of NO in biological systems is nitrite, we measured nitrite content to determine NO levels in tissue lysates from zebrafish embryos and mouse tracheas. The nitrite level was detected by a Sievers Nitric Oxide Analyzer (NOA 208i)[26]. Zebrafish embryos were collected as dry pellet at 8–10 somite stage and chorion was removed with 1 mg/ml pronase. Pools of 50 embryos were used as samples. Mouse tracheas were collected and analyzed as individual samples. Tissues were homogenized in PBS for 1 min on ice with a micropestle, centrifuged for 10 min at 13000 × $g$ at 4 °C and analyzed immediately. Lysate supernatant (10 µl) was separated for BCA assay. Lysate was mixed with cold methanol at a ratio 1:1 and centrifuge at 13000 × $g$ at 4 °C for 15 min to precipitate proteins and assayed in the NOA. Samples (40 µl) were injected into the purging vessel containing triiodide ($I_3$) through the vessel's septa using a Hamilton syringe. Data were collected using Sievers Windows 95/98/NT (version 3.21 PNN) and analyzed by Origin software (Origin 2021) using the areas under the picks to calculate levels of nitrite, normalized by the calibration curve obtained by injecting different doses of nitrite.

## Transmission electron microscopy (TEM)

Embryos at 8–10 somite stage were fixed in 2% glutaraldehyde and 4% PFA in 0.1 M Millonig's buffer, pH7.4, and post-fixed in 1% buffered osmium tetroxide, dehydrated in a graded ethanol series, and embedded in EPON/Araldite. Thin (70 nm) sections were stained with uranyl acetate/lead citrate and photographed with a Hitachi 7650 TEM. We analyzed 12-14 cilia acquired from 4 different embryos per group collected in 4 independent experiments.

## cGMP levels measurement

The amount of cGMP in zebrafish embryos was measured by competitive ELISA assay using Cyclic GMP EIA Kit (Cayman Chemical 581021). To prepare the samples, 50 zebrafish embryos at 10 somite stage were homogenized in 0.1 ml 0.1% hydrochloric acid (HCl) on ice. The supernatant was carefully collected and used for analysis. The protocol was executed following the manufacturer's instructions. Samples were assayed in four dilutions and repeated twice. The multi-well plate loaded with samples and standards was incubated for 18 h at 4 C° and developed with Ellman's Reagent in the dark for 1.5 h at RT. The plate was read at wavelength of 412 nm using a plate reader. The amount of cGMP in each sample was calculated based on the standard curve.

## Protein expression, purification and spectrophotometry

Recombinant zebrafish cytoglobin 1 and 2 and human cytoglobin were expressed in *E. coli* as previously reported[6]. The amino acid sequence corresponding to Nos2b (Uniprot accession A9JNR5_DANRE) was codon optimized for *E. coli* expression, synthesized, and introduced in the pET28a plasmid (Novagen). Plasmid assembly was completed by Genscript (Piscataway. NJ). The plasmid was transformed into SoluBL21 cells (Genlantis). High-level expression of mouse iNOS in *E. coli* requires coexpression with calmodulin, therefore, a plasmid containing human Calmodulin was cotransfected as previously reported for mouse iNOS expression[61]. The purified protein did show spectral properties consistent with the presence of heme and flavins, nevertheless it did not produce the usual 450 nm peak when exposed to CO and dithionite but a 420 nm peak consistent with a five-coordinated ferrous-CO complex. The protein did not show NO synthesis activity either, although it did show CaM-dependent cytochrome *c* reductase activity (data not shown) indicating a functional flavin (reductase) domain. We did not elucidate the causes for the lack of activity, but we speculate that the purified protein did not incorporate properly the tetrahydrobiopterin cofactor. Full length mouse iNOS ($\Delta 65$) and human calmodulin were a kind gift of Dr. Dennis Stuehr (Cleveland Clinic). The reduction of Cygb by NOS was studied by UV-Visible spectroscopy. Spectra were recorded in a Cary50 spectrophotometer. Reaction mixtures included human CYGB (8–10 μM) and 250 μM NADH. In experiments using NOS, either mouse iNOS or zebrafish Nos2b were used (12.5 nM). When CaM was added, a concentration of 12.5 μM was used. Reactions were carried out at 37 °C in HEPES 40 mM, pH 7.6 buffer with 150 mM NaCl.

## SOD activity measurements

The method for assessing SOD activity of Cygb2 was completed as previously described, with slight variations[12]. Bovine xanthine oxidase (XO), xanthine, copper-zinc superoxide dismutase (SOD), and oxidized cytochrome *c* were purchased from Sigma. XO, SOD, and ferric cytochrome *c* were each separately dissolved in PBS and concentration determined by their respective absorbance: XO, at 450 nm ($\varepsilon_{450} = 37.8\ mM^{-1}cm^{-1}$); SOD, at 258 nm ($\varepsilon_{258} = 10.3\ mM^{-1}cm^{-1}$); and cytochrome *c*, at 410 nm ($\varepsilon_{410} = 106\ mM^{-1}cm^{-1}$). Xanthine was prepared freshly in 10 mM sodium hydroxide. All spectra and kinetics were recorded with a Cary 50 spectrophotometer (Agilent) under normal atmosphere. In a cuvette thermostatted at 25 °C, 50 μM ferric cytochrome *c* and 50 μM xanthine were added to PBS and 100 μM EDTA. The reaction was then initiated by adding 70 nM XO, triggering superoxide generation, and kinetics were measured monitoring the Q-band region of cytochrome *c* (450-700 nm). Spectra were taken every 6 s, monitoring the growth of the local maximum absorbance at 550 nm from the resulting ferrocytochrome *c*. Experiments were repeated in the presence of known quantities of the control SOD as SOD lowers the total amount of available superoxide at a given time and thus slows the rate of the reduction of ferricytochrome *c* in a concentration dependent manner. Separately, Cygb2 was added in known concentrations in a similar manner to assess the protein's effect on superoxide production. Initial rates were plotted, and graphs show the mean ± SEM of three separate measurements. Measurements for one entire set (one of each control concentration, one of each zebrafish Cygb concentration) were completed one after another without interruption.

## L-Citrulline assay

The effect of cytoglobin on iNOS activity under hypoxia was determined by citrulline measurement. NOS produces 1 molecule of L-citrulline per molecule of NO produced[62]. L-citrulline, was quantified using the Citrulline assay kit MET-5027 (Cell Biolabs, San Diego, CA) according to manufacturer instructions. Reactions were performed in 40 mM EPPS buffer, pH 7.6 with 150 mM NaCl at 37 °C. Reactions mixtures included 50 nM mouse iNOS, 5 μM human Cygb, 4 μM FAD, 4 μM FMN, 1 μM Human Calmodulin, 0.8 mM $Ca^{2+}$, 0.2 mM EDTA, 100U/ml catalase, and 40U/ml SOD. Control experiments omitted NOS and/or Cygb. The reaction components (120 μl) in aerobic buffer were premixed with 680 μl of anaerobic buffer for a final oxygen concentration of aprox 39 μM $O_2$ (3.2%). Reactions were initiated by the addition of 200 μM NADPH.

## Mouse trachea histology

Mice were euthanized by 100% $CO_2$ exposure and tracheas were isolated through an incision in the neck region. The samples for histology were fixed in 4% paraformaldehyde, dehydrated in ethanol, replaced by xylene and embedded in paraffin. Thin transversal sections (5 μm) were prepared, stained with hematoxylin and eosin (H&E) and analyzed by AxioVision/ Zeiss software. We measured the length of 20 cilia in 6 different areas of each trachea and averaged the values. Equal number of trachea were analyzed for wt and mutants ($n = 4$) collected form 4 independent experiments.

## Reporting summary

Further information on research design is available in the Nature Portfolio Reporting Summary linked to this article.

## Data availability

All data generated or analyzed during this study are included in this published article and its supplementary information files. In addition, data from this study are available from the corresponding authors upon request. Source data are provided with this paper.

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

## Acknowledgements
We thank H. Hammer for zebrafish care, Dr. Dennis Stuehr for the iNOS recombinant protein and Dr. Alexander Krupnick for the human samples. This work was supported by the AHA grant 18CDA34110344 (to PC), the Institute for Transfusion Medicine and the Hemophilia Center of Western Pennsylvania (to M.T.G.), the National Institutes of Health Grants T32 HL110849 (to E.R.), HL157103 (to C.W.L.), P01 HL103455 and T32 HL007563 (to M.T.G.).

## Author contributions
E.R.R. and P.C. designed and conducted experiments and analyzed data; P.C. and M.T.G. wrote and reviewed the manuscript; J.X., M.S., C.S. and A.C. carried out zebrafish experiments; A.H. contributed with providing experimental models; Q.X. and C.L. contributed with planning mouse cilia and laterality experiments; M.T. analyzed and conceptualized zebrafish experiments; A.W.D. and J.T. designed and analyzed biochemical assays. E.R.R., J.T., P.C. and M.T.G. edited the paper.

## Competing interests
The authors declare no competing interests.
