## [Peer Review File · Nature Communications]

REVIEWER COMMENTS

Reviewer #1 (Remarks to the Author):

In this original manuscript entitled “Cytoglobin regulates NO-dependent cilia motility and organ laterality during development”, E. Rochon and colleagues generate and characterize the phenotype of zebrafish lacking *cygb2* function.

The developmental defects are consistent with *cygb2* being required for proper left-right asymmetry establishment by regulating NO-dependent cilia motility and cilia length in the Kupffer vesicle. This is an unexpected and potentially interesting finding; however, more experiments should be performed to answer the following points:

The authors write that the *cygb2* zebrafish mutant model PCR/Kartagener disease. However, they only investigate the LR phenotype/ Kupffer vesicle cilia. To make that claim, the authors must investigate whether the KO embryos also present with additional ciliopathy phenotypes. This involves assessing if motile cilia in the pronephric duct, optic vesicle, brain, olfactory pit are affected as well. If only LR defects are present, then the *cygb2* zebrafish mutant phenocopy heterotaxy, not PCD/Kartagener syndrome.

It is interesting to note that mutant embryos do not show body curvature (a common ciliopathy-related phenotype). Do the *cygb2* mutant fish have normal viability and fertility? Please provide a classical Kaplan-Meier plot of wt, het, hom over several months. What is the phenotype of F3, Maternal and Zygotic KO fish ?

What is the endogenous expression pattern of zebrafish *cygb1*? What is the phenotype of *cygb1* mutants? Could double *cygb1/cygb2* mutants present with more severe phenotypes?

How do the authors explain that only 30% of mutant embryos present with laterality defects (cardiac laterality and expression of LR markers) instead of the expected 50% in case of randomization?

In the RNA scope presented in Figure 2B, *cygb2* expression is asymmetric as it is more expressed on the right side (is it the right side of the embryo? Not indicated). This should be discussed. Is *cygb2* expression downregulated on the left in response to the flow, similarly to *dand5* and *cirop*?

The authors show that injections of cygb2-mRNA in cygb2 801a restored NO/NO-2 to normal levels (Figure 2), however they do not show whether it rescued the LR phenotype?

Which Cygb knock-out mouse line did the authors use (it is not clear in the manuscript)? Is mouse Cygb expressed in the left right organizer?

The authors write “Global knock-out of Cygb in mouse displayed a cilia phenotype similar to zebrafish Cygb2 mutants, with shorter airway cilia and a cilia beat frequency significantly lower in mutants compared to wt” however they do not assess the left-right asymmetry in mouse KO (which is the phenotype they studied in zebrafish)?

Is the function of zebrafish cygb2 in LR establishment conserved in mice? Does the Cygb KO mouse model PCD/Kartagener syndrome, including chronic infections of the respiratory tract, male infertility and LR defects? If not, this should be clearly discussed and the title updated.

Minor points:

In the Western blots shown in Supplementary Figure 1B-C, it is not clear which band represent the cygb2 protein in wt embryos (lanes 1) as there is no band observed at the same size of the recombinant cygb2 protein (lanes 3). While a higher band present in wt embryos (lanes 1) seems absent in mutant embryos (lane 2), it is not clear if this is what the authors are referring to. The western blot should be optimized to obtain publication-quality images.

For cilia length analysis, it would be more relevant to plot the actual cilia length for each cilia analyzed instead of an cilia length average per embryo. Please amend.

The authors should make sure that the left (L) and the right (R) are indicated for each embryo/KV image.

Reviewer #2 (Remarks to the Author):

Elizabeth R.R. et al described the phenotype of cygb2 mutant zebrafish with cardiac and gastro-intestinal tract laterality defects. Although the results seem interesting, however, there many concerns related to this mutant zebrafish and CYGB-KO mice.

1. The method sections do not provide the detail process, especially in animal care section, the authors did not describe how they generate *Cygb* knockout mice. There is only one sentence saying that: "Mouse strains used in this study were wild-type and *Cygb* KO in the C57BL/6 background." Furthermore, there is no information on how they confirm CYGB expression in these mice.

Also in the method sections, *cygb2* mutant generation, there is no detail information on CRISPR-Cas9 genome editing about off-target sites, how they confirm that their designed gRNA have no effect on inducing mutations in other genes.

2. Congenital disorders of ciliary motility are labelled as primary ciliary dyskinesia (PCD). Nearly 50% of PCD patients have situs inversus. Such cases of PCD with situs inversus are known as Kartagener's syndrome (Nat Genet. 2002;30:143–4).

In this study, the authors have no data showing situs inversus in the *cygb2* mutant alleles using CRISPR/Cas9, how can they be described as "modeling Kartagener's syndrome" in the abstract?

3. Primary ciliary dyskinesia (PCD) is an autosomal recessive disorder, in most cases.

Mutations in 39 genes, responsible for an estimated 70% of cases, have been linked to PCD (Hum. Mutat. 2017; 38: 964-969).

Recent recessive loss-of-function mutations in the open-reading frame C11orf70 is reported to cause PCD (Am J Hum Genet. 2018 May 3;102(5):973-984).

Genetic analyses of PCD-affected individuals identified several autosomal-recessive mutations in genes encoding axonemal subunits of the outer dynein arms (ODA) and ODA-docking complexes (Am. J. Hum. Genet. 1999; 65: 1508-1519; Am. J. Hum. Genet. 2008; 83: 547-558; Nat. Genet. 2002; 30: 143-144; Nat. Genet. 2012; 44: 714-719; Proc. Natl. Acad. Sci. USA. 2007; 104: 3336-3341; Nat. Genet. 2013; 45: 262-268, etc.).

Three X-linked PCD variants have been reported so far (Am. J. Hum. Genet. 2017; 100: 160-168; Hum. Genet. 2006; 120: 171-178).

Then, the authors need to verify the presence of these above published genetic mutations related to PCD in their two *cygb2* mutant zebrafish and *Cygb*-KO mice.

4. Supplemental Figure 1 showed that the *cygb2801a* reduced about 50% of CYGB protein expression compared to wt while *cygb2801b* only reduced about 10%. How do these not clear deletions of *Cygb* induce the PCD phenotype?

5. Data from Figure 2 and 5 showed lower levels of Nitrite in the absence of *Cygb* in both zebrafish and mouse, respectively. However these data are in contrast with all publication up to date related to *Cygb*-KO mice which show elevated levels of Nitrite at both physiological and pathological condition in these

mice (Nat Commun 8, 14807 (2017); Sci Rep 6, 24990 (2016); Sci Rep. 2017 Feb 3;7:41888.; Antioxid Redox Signal. 2022 Sep 16. doi: 10.1089/ars.2021.0279)

6. The expression of CYGB in epithelial cells is not generally accepted as almost publications uptodate showing CYGB expression in mesenchymal cells, splanchnic fibroblasts, pericytes. In the lung and heart, its expression in stromal cells (Lab Invest. 2004 Jan;84(1):91-101;Clin Mol Hepatol. 2020 Jul;26(3):280-293). This study used two antibodies anti-CYGB to detect CYGB expression in the mice, however, one of them, are specific for only human tissue (Sigma HPA017757).

Reviewer #3 (Remarks to the Author):

The manuscript by Rochon et al seeks to address an important mystery in LR development, primary ciliary dyskinesia, and NO signaling. For years, clinicians have measured NO levels from the nasally exhaled air of patients and noted that patients with PCD have low levels of exhaled NO compared to healthy patients. This is an important clinical test as patients with PCD do not always have other (more obvious and easily identified) symptoms of PCD such as situs inversus or heterotaxy at presentation. However, they may suffer from respiratory illness or infertility. Given the importance of diagnosing poor respiratory clearance in these patients and providing mucus clearance support/aggressive pulmonary infection treatment, rapid testing via NO has proven highly useful. Given that we simply have no idea how PCD relates to NO signaling, the potential impact of this manuscript is very high. Therefore I am generally enthusiastic of the discovery outlined in this manuscript.

However, in this current form, I cannot recommend acceptance of this manuscript to Nature Communications. Given the high bar and broad audience of Nature Comm, additional mechanistic data is needed to be of interest and impact to the field. Generally, I would reject a paper that required so much mechanistic data in order to be suitable; however, given the potential importance of this paper, I believe this paper should be revised and resubmitted. However, it is not entirely clear at this stage that the authors can identify additional mechanism that is necessary for further insight into NO signaling, cilia structure, and PCD – additional discovery is required.

Currently, the manuscript seeks to understand the role of cytoglobin in embryonic development. This is clearly an unexpected and exciting discovery that would be of interest to the field of NO signaling but not to the broader audience that Nature Comm serves. Previous work has demonstrated that cytoglobin may play a role in NO homeostasis. *cygb2801* mutant embryos have defects in LR. The authors than do a nice job phenotyping and demonstrating that other NO pathway members also give a similar phenotype. However, what is lacking is a mechanism that connects NO signaling to a defect in cilia structure or precisely how NO leads to a lack of motility. I could see a few mechanisms – alterations in notch

signaling that lead to specification of immotile cilia vs. motile cilia, alteration in cilia assembly that lead to loss of the inner double of microtubules, there are certainly others as well. Currently this is just a phenotype associated with the mutant embryos. Unfortunately, without some insight into the mechanism directly connecting NO signaling and cilia function the paper falls short of the bar for Nature Comm.

Concerns:

1. The paper seems to begin with cytoglobin as the discovery. This reduces the impact of the paper – really this should be about NO signaling and LR patterning. Cytoglobin is an exciting avenue that leads to this but really the impact of the paper is on NO signaling and mechanisms of PCD.
2. There is a brief description that somites and craniofacial development are abnormal in *cygb2801* mutant embryos. Does this phenotype help identify potential mechanisms? The craniofacial cartilages are products of an interaction with the neural crest and the surrounding mesenchyme and requires a series of interactions with different signaling processes (Wnt, BMP, etc). Could this be an avenue for mechanism discovery that could also be applied for LR patterning? I am not suggesting that the authors expand work on craniofacial or somite patterning unless it informs their work on LR. In fact, I would suggest that they remove craniofacial patterning and somite analysis unless they can link it to the underlying signaling mechanism – I believe the authors are trying to be thorough (which is laudible), but the manuscript is focused on LR patterning which is enough.
3. There is also a comment on shortened body axis. Looking at the fish it appears that they may have scoliosis(?). Scoliosis has been associated with cilia dysfunction in fish and if this is the case might be worth discussing especially if it is informative to the LR story.
4. Fig 2 – the authors state that *Cygb2* is expressed in the cilia. Frankly the relatively low mag imaging is not entirely clear. A high magnification would help a lot in convincing the reader. Additionally, the imaging should be done in the *cygb2801* mutants to demonstrate the signal is lost and therefore specific. There is well known false positive where primary antibodies mixed with acetylated tubulin primary antibody lead to cilia signal that is false. I just realized that they have immunolocalization in figure S3a in the context of control MO and *cygb2* MO injected embryos. Looks like the signal is reduced but hard to tell at this low magnification. This could help. Is the antibody used for western blot the same as the immunolocalization?
5. More importantly, if cilia localization of *cygb2* is true, what does it mean? Does this mean that cytoglobin is affecting NO signaling in the cilium itself? How does that affect cilia structure given that the cilia appear smaller? The authors do not discuss intraciliary NO signaling and what this might mean? This is critical for understanding mechanism.
6. Fig 2 – Amazingly, *cygb2* appears to be localized to the right of the LRO (left right organizer) (I think – the images in 2b lack orientation of what is anterior-posterior/left-right – this needs to be corrected).. Is *cygb2* expressed symmetrically at earlier stages? For example *dand5* is expressed symmetrically at earlier stages and then becomes asymmetric at later stages (*dand5* becomes right sided with reduced expression on the left due to LRO flow). Is this also true for *cygb2*? This would be utterly fascinating and

would suggest that NO signaling is also lateralized. If so this would be downstream of cilia signaling. This needs to be addressed.

7. Examination of the expression of *cygb2* is essential at earlier stages. Given the asymmetric expression of *cyb2* at 10 somites how do the authors explain the cilia expression? As above, is the expression symmetric across the LRO or not at earlier stages that then become asymmetric? Is it responsive to LRO flow or not – please take a look at martin blum's paper in *Xenopus* about *dand5* (aka *coco*). Also the recent paper by Yuan in *Science* also takes advantage of *dand5* in zebrafish. Also looking at *nos2b* – it would be interesting to examine this gene's mRNA expression at earlier stages as well.

8. The LRO has two types of cilia - motile and immotile cilia. As the authors suggest manipulation of NO signaling appears to shorten the cilia – this could represent a transformation of the cilia from a motile cilium into an immotile cilium. This has previously been implicated by Notch signaling (see the papers by Lopes lab and Khokha lab). Therefore, one explanation for the change of length could be a change in notch signaling and a change in cilia type. Another possibility is that as the authors seem to suggest there is a change in cilia structure. It is a bit tough to tell what is going on with the quality of the TEM – perhaps the loss of the central pair is a change from cilia type (ie transformation to an immotile cilium). Unfortunately the TEM is not of very high quality. Certainly getting good TEM of the LRO is technically challenging. More importantly how does one connect a reduction in NO signaling to this change in cilia structure?

9. In Fig 2, the authors nicely test that mutants of *cygb2* have reduced levels of NO and that this can be rescued by the expression of *cygb2* mRNA. This leads to a number of questions: 1) does the gain of NO by GOF of *cygb2* affect LR development or development in general – this might give clues to its function 2) does other models of PCD cause decreased levels of NO – for example, *dnah9* depletion or *c21ORF59* depletion? What about loss of cilia? This is relevant since patients with PCD also have low levels of NO and the authors are supremely positioned to test if they can make zebrafish a model for testing levels of NO and understanding mechanism. This could create a whole new avenue and answer a critical question in the field – why do PCD patients have lower NO levels? Additionally, they could test if adding NO agonists in that context, rescues the PCD phenotype! This might be exceptionally impactful.

10. Fig 4E – this is confusing. The authors state that DETA/NO treatment completely rescued laterality defects in mutants at a concentration of 250 μ M – effect is dose and stage specific when embryos are treated prior to gastrulation – however according to the time chart the embryos treated at 2-4 cells and oblong are before gastrulation and have cardiac looping defects similar to untreated mutants while those treated later are rescued – dome and epiboly? This doesn't make any sense. No statistical testing is done.

11. Mouse work – Fig 5A – this is not helpful. We need a higher magnification image of the A to be able to see the cilia.

Minor Concerns:

1. Two alleles generated from the same set of two guide RNAs were used to create the mutants used in this study. Ideally, the two alleles would be generated from guide RNAs that do not overlap. Unlike random mutagenesis, CRISPR may hit the same off target so rather than make two alleles from the same

sgRNAs it would be preferable to make a second allele with non-overlapping sgRNAs. To be clear, reduction of protein expression demonstrates that the LOF alleles are effective but not specific – there may still be a second site that is mutated that is causing the phenotype.

2. Western blot – Fig S1 – better annotation is needed. Clearly the antibody detects a couple of bands in fish – I believe the band that the authors wish to highlight which is missing in *cygb2801a* is running a bit higher than the *Cygb2* recombinant protein. Of note this band remains present in the *cygb2801b* allele which should predict a much more severe truncation? What is going on here?

3. SFig 4 - no statistics tested in the graphs. This needs to be corrected. Is there synergy between *Nos2b* and *cygb2*? In SFig4a – looks like the incidence of the cardiac loop is lower in the *cygb2801a* allele when *nos2b* MO is applied – does this make sense?

4. SFig5 – again no statistics in any of the graphs.

5. The heart is normally on the left. But the cardiac loop goes to the right (ie dextral looping) this is the standard nomenclature in the field. Left heart loop would be the reverse but it seems the authors are not using this in the conventional way – Fig 4F,G for example – the heart should be on the left and the cardiac loop should be rightward. This is pretty basic in the field – might be of some use to get a LR expert to review some of these data to make sure the nomenclature is right – otherwise this is pretty confusing.

6. A number of figures could benefit from additional labeling. Many times it is difficult to follow the figures without reading the figure legends when just some simple labels to the figures would improve readability considerably.

Reviewer #4 (Remarks to the Author):

This is a novel manuscript that reports that Cytoglobin 2 (*Cygb2*) in Zebra fish regulates ciliary motility and organ laterality. This is new, important, and unexpected as the authors note. It is clearly shown that genetic deletion of zebra fish *Cygb2* contributes to cardiac and GI tract laterality defects. In zebra fish *Cygb2* colocalizes with *NOS2b* and modulates NO sGC signaling with this affecting ciliogenesis and cilia motility that is required for left-right patterning in zebra fish. This effect on modulating embryogenesis is new and very interesting.

The manuscript presents a large amount of data and attempts to span implications from Zebrafish to mice to man. These efforts to seemly over extend the implications of the work is a weakness of an otherwise largely solid and strong paper. While *Cygb* deficiency in Zebra fish is compared to Kartagener's Syndrome in humans, this seems valid in some ways including situs inversus and ciliary dysfunction but perhaps not in others such as sinusitis, bronchiectasis. In mice comparison is made to *Cygb*^{-/-} mice where ciliary dysfunction is seen but these mice do not have situs inversus but were found to have ciliary dysfunction in the current study. One can not assume or expect that the properties seen in zebra fish precisely extend to mice and man. The properties of *Cygb2* in zebra fish may be somewhat different

than Cygb in mammalian systems. In fact there are two Cygb isoforms in zebra fish and here only Cygb2 is studied as it has 75% homology to mammalian Cygb, while Cygb1 has 63% homology. Only Cygb2 is studied here with CRISPER/Cas9 knockout; however, the role and properties of Cygb1 are not studied or commented on.

It is stated several times that the mechanisms underlying Cygb function are unknown (as in line 77) and that its physiological function is unknown (lines 33, 241). This is inconsistent with the prior work of this group and others in the field. Indeed the paper properly presents the functions identified to date, as a NO dioxygenase, nitrite reductase, redox sensor, and antioxidant / SOD function. The role of Cygb in embryogenesis, development, and ciliary function is new and important; however, clearly there is much prior data on the role and functions of Cygb. As such the authors would be best to reword these lines to focus on what is new here without detracting from what is already known and established.

There are several specific concerns that should be addressed by the authors as listed below.

1. Line 33 “unknown physiological function” this should be reworded, such as “questions remain regarding its functions. Its role in development and embryogenesis is unknown”
2. The paper focuses on Cygb2. Was Cygb1 also measured? Does it also localize with cilia? Where does it localize?
3. On lines 69-71 Again, it is suggested to simply say that questions remain regarding its precise range of functions and its role in development has not been studied.
4. On line 79 here or in the discussion, it would be good to comment on similarities and differences in structure compared to human Cygb particularly in the vicinity of the heme.
5. Line 88, with regard to the organ laterality why is this only 87.3% in control? Is this expected? In the discussion, it would be good to discuss what a given % change implies, that if Cygb2 was essential for laterality this would decrease to 0, so it seems to contribute to this process of development but is not essential for it.
6. Line 88-89, this data as stated does not seem to be consistent with supplement figure 1D, please check this and explain this.
7. Lines 135-145 and figure 2 F & G, there are several concerns here and a need for additional information. First of all, NO can be oxidized to nitrite or nitrate, with nitrate usually the major product due to NO dioxygenation by heme proteins such as Cygb. Since Cygb is the focus of the paper, it seems odd to ignore this. As such, it would be logical to measure both. Here only nitrite is measured. Alternatively, one could directly measure NO. There are also technical questions and concerns regarding the nitrite/NO measurements of Figure 2. In Fig 2 G the units are shown as nM/ug, it seems that ug is the protein concentration but this is not defined here. The protein concentration is specified on the x-axis as 50 or 100 pg. It is unclear what “co” is as this is not defined. Please define what “co” is. With 50 pg or 100 pg of protein with the measured nitrite of 0.1 to 0.3 nM/ug protein, the levels would be well below the sensitivity detection limit of this technique. Please check this figure and explain this. In general the detection sensitivity limit of this method even with larger volumes is about 10 nM at best. Another related problem is that typical buffers can contain 0.1 to 1 uM amounts of nitrite and this

background level would well exceed the levels noted. Again please clarify the conditions here including the protein concentration, the volumes used and background nitrite levels. Also for the WT, the observation number of 3 or 4 points for 50 and 100 pg seems insufficient in view of the variability.

8. Line 169-170 and lines 179-180, while Nos2b knock-down phenocopies the Cygb mutant, this does not prove a direct Cygb-Nos interaction. As noted in the paper and shown by this group and others, Cygb can be a potent nitrite reductase under hypoxic conditions, as Nos is a major source of nitrite, it could be required for NO generation by Cygb2. That is there could be 2 parallel pathways of NO generation with Cygb requiring NOS as a source of nitrite.

9. What is the oxygen level in these zebra fish embryos? Measurement of this would be helpful to understand how Cygb2 is functioning here.

10. Line 185 the spelling error “suggestiong” should be corrected to “suggesting”

11. Supplemental figure 5 B, 0 nM SOD should be 0 Cygb2. Interestingly, it appears in 5C that SOD+catalase decreased laterality. Is this so?

12. Discussion, Line 241 Again suggest to change this wording

13. Where Kartagener’s syndrome is noted, it would be good to note similarities and differences and what is known about the molecular basis of this disorder. Do the authors intend to suggest that there is a defect in Cygb expression or structure in these patients?

14. Line 247 In view of the prior work from the authors and others in the field on the potent NO dioxygenase function of Cygb, how do the authors explain the lower levels of NO and nitrite reported. As in the presence of oxygen Cygb will efficiently scavenge NO and likely nitrite will be depleted as well, how can this be explained? Is it proposed that the embryos are markedly hypoxic? If so Cygb would be expected to function as a nitrite reductase?

15. Line 293 regarding Cygb modulating NO production in cilia through a NOS-dependent process, so far there is no established basis for a direct Cygb NOS2 interaction. It is more likely that this is due to NOS2 being a major source of nitrite, but the NO production would require a low oxygen tension. Perhaps this exists in the zebra fish embryos. It would be interesting to see if one adds nitrite with the nos2b knock down if this can restore NO generation and laterality.

16. Lines 305 and 320 It is not clear that Cygb is modulating NOS function, it seems more likely that NOS is modulating Cygb-mediated NO production. All of these processes both Cygb function and NOS function are oxygen-dependent, thus there is a critical need to measure the oxygen levels in this system to understand the underlying regulation of NO generation.

17. In spite of the current limitations in mechanistic understanding noted above, the fundamental effects of Cygb2b knockout on embryogenesis are of high importance and novelty in themselves. Thus, addressing and considering the above comments may be sufficient for the current manuscript as there are limits as to what amount of data can be expected in one manuscript and the paper already seems somewhat overextended. It is simply important to address the technical concerns and the conceptual issues noted above.

REVIEWER COMMENTS

Reviewer #1 (Remarks to the Author):

We appreciate the expert comments provided by reviewer 1, which also align with suggestions from other reviewers. We have performed extensive new experiments to address these suggestions, including studies with cytoglobin 1 (*cygb1*) knock-outs (to test for upregulation of the other isoform), new antibody detection and validation of *cygb2*, and Kaplan-Meier survival studies.

In this original manuscript entitled “Cytoglobin regulates NO-dependent cilia motility and organ laterality during development”, E. Rochon and colleagues generate and characterize the phenotype of zebrafish lacking *cygb2* function.

The developmental defects are consistent with *cygb2* being required for proper left-right asymmetry establishment by regulating NO-dependent cilia motility and cilia length in the Kupffer vesicle. This is an unexpected and potentially interesting finding; however, more experiments should be performed to answer the following points:

The authors write that the *cygb2* zebrafish mutant model PCR/Kartagener disease. However, they only investigate the LR phenotype/ Kupffer vesicle cilia. To make that claim, the authors must investigate whether the KO embryos also present with additional ciliopathy phenotypes. This involves assessing if motile cilia in the pronephric duct, optic vesicle, brain, olfactory pit are affected as well. If only LR defects are present, then the *cygb2* zebrafish mutant phenocopy heterotaxy, not PCD/Kartagener syndrome. It is interesting to note that mutant embryos do not show body curvature (a common ciliopathy-related phenotype).

We thank the reviewer for these comments. We received similar feedback from reviewer 2, asking us to focus on the findings of left-right patterning: “I believe the authors are trying to be thorough (which is laudable), but the manuscript is focused on LR patterning which is enough”. In order to address these common concerns we have performed additional analysis of the zebrafish otolith formation and otic vesicle by comparing *wt* and *cygb2* mutants during development. We did not find any obvious defects. Additionally, as suggested by reviewer 3, “looking at the fish (Fig. Suppl. 1E) it appears that they may have scoliosis (?). Scoliosis has been associated with cilia dysfunction in fish”, we added the analysis of body curvature in adult *cygb2* mutants that may indicate scoliosis. In this case, as the reviewer suspected, we found a significant number of fish with mild but obvious body curvature, and we added this analysis in new Fig. S1F. In order to be more conservative in our interpretation of these data we have rephrased the text (lines 36-37, 273-275) to focus on our primary findings on left-right determination, as requested by reviewer 2:

Do the *cygb2* mutant fish have normal viability and fertility? Please provide a classical Kaplan-Meier plot of *wt*, *het*, *hom* over several months. What is the phenotype of F3, Maternal and Zygotic KO fish ?

We thank the reviewer for pointing this out. Although in the limited time given by the review process, we could not analyze survival trends after several months, we were able to analyze up to 50 days post fertilization survival rates and we added the Kaplan-Meier curves in revised Fig. S1H. Importantly, we

added the data analysis on our new *Cygb1* mutants. No statistically significant differences were observed among the different groups. All the experiments were carried in F3/F4 maternal zygotic knockout zebrafish. Our findings suggest a hypomorphic cilia dysfunction phenotype, as compared to complete cilia knock-out, reflecting the finding that cytoglobin-NOS is an up-stream positive modulator of cilia function. This conclusions are added in the text (lines 299-301).

What is the endogenous expression pattern of zebrafish *cygb1*? What is the phenotype of *cygb1* mutants? Could double *cygb1/cygb2* mutants present with more severe phenotypes?

To detect possible redundancies or compensations between *cygb2* and *cygb1* we generated the *cygb1* mutants (*cygb1⁸⁰²*) and analyzed the left-right patterning in development. We added this analysis in Fig. S1I and in the the Results (lines 107-111):

“Because of the high similarity between *Cygb2* and *Cygb1*, we reasoned that possible compensatory mechanisms could be at play in the *cygb2^{801a}* knock-out. Using CRISPR, we generated a *cygb1^{802a}* mutant maternal zygotic line lacking *Cygb1* protein (Fig. S1B, S1G) and analyzed survival rates and cardiac laterality. No significant defects in survival and cardiac left-right patterning were detectable (Fig. S1 H-I).”

We also observed normal viability in F3 maternal zygotic *cygb1⁸⁰²* knockouts by Kaplan-Meier analysis (Fig. S1H). We analyzed *Cygb1* protein expression by western blot to confirm the expression of *Cygb1* in *cygb2* mutants similar to wt and the lack of *Cygb1* in the *cygb1* mutants as expected (new Fig S1B).

How do the authors explain that only 30% of mutant embryos present with laterality defects (cardiac laterality and expression of LR markers) instead of the expected 50% in case of randomization?

Our data define *Cygb2* as upstream positive modulator of cilia function through NOS-NO-sGC-cGMP signaling pathway. Similar to human subjects with a “mild or hypomorphic” phenotype, these animals present with a phenotype not as severe as direct loss of function mutations in cilia. We would like to point out other reports where cilia function ablation does not cause the expected 50% laterality distribution, indicating that compensatory mechanisms could be at play (Pintado et al. R Soc Open Sci. 2017 Mar 8;4(3):161102; Li et al. PLoS Genet. 2016 Feb 26;12(2):e1005821).

In the RNA scope presented in Figure 2B, *cybg2* expression is asymmetric as it is more expressed on the right side (is it the right side of the embryo? Not indicated). This should be discussed. Is *cybg2* expression downregulated on the left in response to the flow, similarly to *dand5* and *cirop*?

This is a very intriguing observation by the reviewer. First, we added the left-right labeling to the revised figures to help clarify this aspect. Second, we explored the possibility that *cygb2* expression in the KV may be downregulated on the left in response to flow as suggested by the reviewer. Although some

differences are noted in the figure, statistical analysis of all our RNAscope data did not detect statistically significant differences between left and right expression across all experiments.

The authors show that injections of *cygb2*-mRNA in *cygb2* 801a restored NO/NO₂⁻ to normal levels (Figure 2), however they do not show whether it rescued the LR phenotype?

To provide a complete assessment of the phenotype rescue experiment we analyzed cardiac laterality in *cygb2*-mRNA injected embryos (100 pg). Analysis by Chi-squared test show no significant differences among the two groups (Fig. R1). Since the effect of injected mRNA on cardiac laterality are observed much later (2 days post fertilization) compared to the time of injection at 1-cell stage, it is possible that the mRNA is degraded by the time the cardiac left-right orientation is established. However, upon *cygb2*-mRNA injections the consequent effect of increasing NO may still be detectable earlier at the beginning of gastrulation, the time at which we measured NO levels. We have added this to the results (lines 152-155): “Injections of *cygb2*-mRNA in *cygb2*^{801a} mutants restored NO/NO₂⁻ to normal levels at the early gastrula stage (Fig. 2 E-G) but did not rescue the laterality phenotype observed at 2 dpf, possibly due to the mRNA degradation upon injection over the longer period of time necessary to determine correct laterality. “

Figure R1: Percentage of embryos with right/bilateral or left sided heart in *cygb2*^{801a} and *cygb2*^{801a}-mRNA injected embryos. Number of embryos is shown above the graph in red. Chi-Squared test, ns, not significant.

Which *Cybg* knock-out mouse line did the authors use (it is not clear in the manuscript)? Is mouse *Cybg* expressed in the left right organizer? The authors write “Global knock-out of *Cybg* in mouse displayed a cilia phenotype similar to zebrafish *Cygb2* mutants, with shorter airway cilia and a cilia beat frequency significantly lower in mutants compared to wt” however they do not assess the left-right asymmetry in mouse KO (which is the phenotype they studied in zebrafish)? Is the function of zebrafish *cygb2* in LR

establishment conserved in mice? Does the *Cybg* KO mouse model PCD/Kartagener syndrome, including chronic infections of the respiratory tract, male infertility and LR defects? If not, this should be clearly discussed and the title updated.

We received the *Cybg* knock-out mouse as a kind gift from Dr. Anders Hay-Schmidt, University of Copenhagen. His lab developed a constitutive KO allele from a *Cygb* conditional KO mouse via deletion of exon 2. The experimental procedure was published in Yassin et al., Sci Rep. 2018 May 2;8(1):6905. We included this information in the methods (lines 367-370).

We agree with the reviewer that a detailed analysis of the mouse knock-out phenotype would clarify whether *Cygb* function is conserved. We also have responded to reviewers

request to focus the manuscript on zebrafish models and mechanisms of cytoglobin-NOS activation and role of NO-sGC-cGMP signaling. We do provide here for reviewers our more preliminary data highlighting effects observed in mouse models. These data require more robust studies beyond the focus of this first report, but can reassure the editors and reviewers that these findings are important and translatable beyond zebrafish.

The association between PCD and congenital heart disease is due to the common requirement of motile cilia in airway clearance and embryonic left-right patterning. This is evidenced by $\approx 50\%$ of PCD patients showing heterotaxy or *situs inversus* (Kartagener syndrome). Cardiac abnormalities including atrial and ventricular septal defects are also often associated with *situs inversus* (Klena NT, Gibbs BC and Lo CW. *Cold Spring Harb Perspect Biol.* 2017;9). Using in utero ultrasound and Doppler echocardiography we analyzed 8 pregnant wt females and 16 pregnant *Cygb*^{-/-} females, crossed with *Cygb*^{-/-} males at embryonic stage E16.5. The total number of mutant fetuses was 101. To obtain the fetal survival percentage (Fig. R2) we compared the number of fetuses in utero at E16.5 and the number of newborn pups in wt and mutants. We detected a 37% decrease in survival of homozygous mutants compared to wt. 2% of mutant fetuses showed *situs inversus* and mesocardia and 4.9% had ventricular septum defects (VSD), consistent with some of the cardiac clinical manifestations of PCD (Fig. R2). Additionally, we observed 4.9% of fetuses with diffused hygroma and pericardial fluid accumulation and 18.9% with renal pyelectasis, which is also observed in patients with ciliopathies in association with congenital heart laterality defects (Gabriel et al. *Frontiers in pediatrics.* 2018;6:175). These data point at the occurrence of embryonic malformations, which reduce viability, and evidence of cardiac developmental defects similar to those reported in PCD. Since we have yet to identify the exact nature of these malformations and more extensive experiments are needed, as well as pathological validation in more fetuses, in the present manuscript we focus on mouse airway cilia expression and localization, as well as effects on NO formation. A complete analysis of the mouse embryonic development is beyond the scope of the current manuscript and we plan to further investigate these developmental changes in future work. This also aligns with requests by other reviewers to focus this manuscript on zebrafish development and mechanisms of NOS activation.

Minor points:

In the Western blots shown in Supplementary Figure 1B-C, it is not clear which band represent the *cygb2* protein in wt embryos (lanes 1) as there is no band observed at the same size of the recombinant *cygb2* protein (lanes 3). While a higher band present in wt embryos (lanes 1) seems absent in mutant embryos (lane 2), it is not clear if this is what the authors are referring to. The western blot should be optimized to obtain publication-quality images.

In the previous supplementary figure 1 we used the recombinant *cygb2* (22kDa) for comparison with the wt sample showing the band at 21kDa. The difference in size is given by the Histidine-tag added to the recombinant protein for expression and purification purposes, which adds 19 extra aminoacids (2.1 kDa). As this may lead to confusion, we repeated the western blot analysis to include a better-quality image in the revised manuscript. We report now the analysis of our two *cygb2* mutant lines *cygb2*^{801a} and *cygb2*^{80ba} and added the analysis of our new *cygb1*⁸⁰² mutant (new Fig S1B). Custom antibodies made against *cygb1* and *cygb2* were used. The band corresponding to *cygb2* is detectable in wt and *cygb1* mutants but lacking in the two *cygb2* mutant samples. Similarly, the band corresponding to *cygb1* is detectable in the two *cygb2* mutant lanes and wt, but lacking in the *cygb1*⁸⁰² mutant samples. We have added this new information in revised Fig. S1B.

For cilia length analysis, it would be more relevant to plot the actual cilia length for each cilia analyzed instead of an cilia length average per embryo. Please amend.

We chose to represent cilia length as average per individual rather than each one cilium length since this analysis reflects the actual biological repetitions. We measured the length of at least 15 cilia per zebrafish embryo, over a total of 10-12 embryos per group in each experiment.

The authors should make sure that the left (L) and the right (R) are indicated for each embryo/KV image.

We have added this accordingly.

Reviewer #2 (Remarks to the Author):

We appreciate the important constructive suggestions from reviewer 2 and have responded to all with new experimental data as well as methodological and editorial clarifications.

The method sections do not provide the detail process, especially in animal care section, the authors did not describe how they generate *Cygb* knockout mice. There is only one sentence saying that: “Mouse strains used in this study were wild-type and *Cygb* KO in the C57BL/6 background.” Furthermore, there is no information on how they confirm *CYGB* expression in these mice.

We thank the reviewer for bringing up this important point. This was raised by reviewer 1 as well and we detailed our answer above. Briefly the *Cygb* knock-out mouse was a gift from Dr. Anders Hay-Schmidt, who developed a constitutive KO allele in the C57BL/6J background mouse. The experimental procedure was published in Yassin et al., *Sci Rep.* 2018 May 2;8(1):6905. In the revised paper we added this information. In addition, we analyzed *Cygb* expression in *Cygb*^{-/-}, *Cygb*^{+/-} and *Cygb*^{+/+} mouse trachea homogenate and verified the lack of *Cygb* band around 21 kDa in the KO lane, as expected. Of note, the recombinant protein (r*Cygb*) is bigger (~ 23 KDa) compared to the *Cygb* of trachea samples due to the His-tag addition to the recombinant protein. We added this analysis in revised Fig. S7B.

Also in the method sections, *cygb2* mutant generation, there is no detail information on CRISPR-Cas9 genome editing about off-target sites, how they confirm that their designed gRNA have no effect on inducing mutations in other genes.

We performed mutagenesis in zebrafish using standardized methods for CRISPR-Cas9 genome editing as described previously (Vejnar et al. *Cold Spring Harb Protoc.* 2016 Oct 3;2016(10); Gagnon et al. *PLoS One.* 2014 May 29;9(5):e98186; Cong et al. *Science.* 2013 Feb 15;339(6121):819-23; Jao et al. *Proc Natl Acad Sci U S A.* 2013 Aug 20;110(34):13904-9). The following information has been added to our supplemental methods section (lines 374-381).

“We injected two gRNAs targeting different sequences in exon 1. Sequences of gRNA primers and of primers for screening mutations are listed in table S1. We recovered two F1 adult fish carrying a 4 bp and a 1 bp deletion with frame-shift mutations and established two stable lines, *cygb*^{801a} and *cygb2*^{801b} respectively. The gRNAs were designed using online software (Montague et al. *Nucleic Acids Res.* 2014 Jul;42(Web Server issue): W401-7) to possibly prevent off-target effects. The F1 adult fish carrying the recovered alleles and the ones showed to be wild-type were outcrossed with uninjected fish, the progeny

was screened for the mutation of interest and the mutant fish were bred to F3/F4 generation before using them for experiments. The wild-type controls are siblings selected at the time of the recovered mutation.“

The laterality phenotype remained consistently detectable in the two *cygb2* alleles and was fully phenocopyed by morpholino knockdown experiments.

2. Congenital disorders of ciliary motility are labelled as primary ciliary dyskinesia (PCD). Nearly 50% of PCD patients have situs inversus. Such cases of PCD with situs inversus are known as Kartagener's syndrome (Nat Genet. 2002;30:143–4).

In this study, the authors have no data showing situs inversus in the *cygb2* mutant alleles using CRISPR/Cas9, how can they be described as “modeling Kartagener’s syndrome” in the abstract?

Thank you for asking for this clarification. We have clearly demonstrated that zebrafish *Cygb2* depletion causes cardiac laterality defects in two mutant lines and in the complementary morpholino knockdown models, consistent with Kartagener’s syndrome. We found abnormalities in early markers of left-right patterning (Fig. 1E-F), and we were able to reverse the laterality defects with an NO donor, a sGC activator, and a *Nos2b*-mRNA construct. In regard to the mouse *Cygb* KO phenotype, we agree that we have not shown laterality defects in this manuscript. As this point was brought up by reviewer 1 as well, we have added here a new analysis by in utero ultrasound looking at the fetal development at embryonic stage E16.5. The data are reported above (Fig. R2). We detected situs inversus and mesocardia at very low incidences, ventricular septum defects (VSD), diffused hygroma and pericardial fluid accumulation and renal pyelectasis, which is also observed in patients with ciliopathies in association with congenital heart laterality defects (Gabriel et al. *Frontiers in pediatrics*. 2018;6:175). Since we are still in the process of characterizing these malformations by ultrasounds and pathological analysis in larger numbers, we will be more circumspect in our interpretation and indicate that these effects reflect ciliary dysfunction and a laterality defect, but not clearly modeling Kartagener’s syndrome. We rephrased the abstract and the text to more conservatively focus on left-right cardiac determination in zebrafish and cilia function (lines 36-37, 273-275).

3. Primary ciliary dyskinesia (PCD) is an autosomal recessive disorder, in most cases.

Mutations in 39 genes, responsible for an estimated 70% of cases, have been linked to PCD (Hum. Mutat. 2017; 38: 964-969). Recent recessive loss-of-function mutations in the open-reading frame *C11orf70* is reported to cause PCD (Am J Hum Genet. 2018 May 3;102(5):973-984). Genetic analyses of PCD-affected individuals identified several autosomal-recessive mutations in genes encoding axonemal subunits of the outer dynein arms (ODA) and ODA-docking complexes (Am. J. Hum. Genet. 1999; 65: 1508-1519; Am. J. Hum. Genet. 2008; 83: 547-558; Nat. Genet. 2002; 30: 143-144; Nat. Genet. 2012; 44: 714-719; Proc. Natl. Acad. Sci. USA. 2007; 104: 3336-3341; Nat. Genet. 2013; 45: 262-268, etc.).

Three X-linked PCD variants have been reported so far (Am. J. Hum. Genet. 2017; 100: 160-168; Hum. Genet. 2006; 120: 171-178).

Then, the authors need to verify the presence of these above published genetic mutations related to PCD in their two *cygb2* mutant zebrafish and *Cygb*-KO mice.

We apologize for any confusion, but we are not invoking cilia mutations in our cytoglobin KO models. Our data suggests that *Cygb* is an upstream positive regulator of cilia function through activation of the

NOS-NO-sGC-cGMP signaling pathway. This is very novel as *a globin has not been shown to positively regulate NOS or cilia*.

We would like to point out that our background animals do not carry any PCD mutations. We would also note that some of the large scale mutagenesis animal studies have linked *Cygb* to hypoplastic left heart syndrome (HLHS) (Liu et al., Nat Genet. 2017 Jul;49(7):1152-1159). Because cytoglobin is an upstream positive regulator of cilia structure and function, it is possible that a primary mutation in cilia genes may be associated with compensatory upregulation of the CYGB-NOS-NO-sGC pathway. Testing some of the PCD genetic variants, particularly the ones responsible for “hypomorphic” phenotypes, is an attractive strategy that would allow to determine the impact of these variants on CYGB-NOS-NO-sGC signaling pathway and to assess whether upstream therapeutic activation of cytoglobin, NOS, NO, or sGC can improve cilia function in these human PCD “hypomorphic” mutations. While beyond the scope of our current paper, these studies will be the focus of future work in our lab.

We have added these important clarifications to our discussion in paragraph 2 (lines 298-300).

4. Supplemental Figure 1 showed that the *cygb2801a* reduced about 50% of CYGB protein expression compared to wt while *cygb2801b* only reduced about 10%. How do these not clear deletions of *Cygb* induce the PCD phenotype?

We noticed that the western blots shown were confusing as the molecular weight corresponding to each *Cygb* was not very clear. The recombinant protein used as control includes an extra 2 kDa due to the Histidine tag which also added to the confusion. We have cropped the blot to show the pertinent molecular weight. We also repeated the analysis of *Cygb2* expression in wt and *Cygb2* knockout in addition to *Cygb1* knockout. We report now the analysis of our two *cygb2* mutant line *cygb2^{801a}* and *cygb2^{80ba}* and added the analysis of our new *cygb1⁸⁰²* mutant (new Fig. S1B). The band corresponding to *cygb2* is detectable in wt and *cygb1* mutants but absent in the two *cygb2* mutant samples. Similarly, the band corresponding to *cygb1* is detectable in the two *cygb2* mutant lanes and wt, but lacking in the *cygb1⁸⁰²* mutant sample. We have added this new information in the revised Fig. S1B.

5. Data from Figure 2 and 5 showed lower levels of Nitrite in the absence of *Cygb* in both zebrafish and mouse, respectively. However these data are in contrast with all publication up to date related to *Cygb*-KO mice which show elevated levels of Nitrite at both physiological and pathological condition in these mice (Nat Commun 8, 14807 (2017); Sci Rep 6, 24990 (2016); Sci Rep. 2017 Feb 3;7:41888.; Antioxid Redox Signal. 2022 Sep 16. doi: 10.1089/ars.2021.0279).

The reviewer refers to data related to intravascular NO production, where cytoglobin has been shown to scavenge NO via the dioxygenation reaction. In our studies we are evaluating zebrafish embryo lysates and trachea lysates, where our results suggest that cytoglobin positively modulates *Nos2b* (in zebrafish) and epithelial inducible NOS (iNOS) (in mice), respectively. These studies in epithelial cells have never been performed previously. We have now clarified in the discussion that all the previous data about *Cygb* possible roles in NO metabolism were collected in smooth muscle cells (Liu et al., Nat Comm. 2017, 8, 14807), and hepatic stellate cells (Sci Rep. 2016, 6, 24990); Van Thuy et al. Sci Rep. 2017 Feb 3;7:41888; Okina et al., Antioxid Redox Signal. 2023 Mar;38(7-9):463-479.).

Available data indicates that Cytoglobin can *directly* regulate NO levels in two opposing ways: *i*) NO generation: reduced CYGB (Fe²⁺) can directly produce NO by nitrite (NO₂⁻) reduction, and *ii*) NO depletion: O₂-bound Cygb reacts at diffusion-limited rates with NO (NO dioxygenation) producing nitrate (NO₃⁻) as final species. Our data supports a *iii*) *novel, indirect mechanism* where Cygb either interacts with, and stimulates NOS-dependent NO formation from L-arginine oxidation by improving oxygen delivery to NOS, or uses NOS as a reductase to support NO generation from nitrite. NO levels are tightly regulated in cilia, and Cygb could work in a concerted manner with NOS by the mechanisms described above, to finely tune NO levels to the optimal values for cilia function. In order to dissect the relevance of the three mechanisms described, we used the NO analyzer (NOA) for gas phase chemiluminescence assays optimized in our lab, we screened NO levels in our *Cygb* mutant models. We found higher levels of nitrite/NO formation in both zebrafish embryos and mouse tracheas from wt compared to *Cygb* KO animals. Given the lower NO levels our data suggest a role for Cygb in stimulating NO production. The specific mechanism at play is not yet clear, but our new data suggests that Cygb may serve as an oxygen carrier providing oxygen for NOS function at low oxygen tensions. We added the analysis in the revised manuscript (new fig 5G). Importantly, these data are also supported by the data showing a decrease in cGMP levels in zebrafish *cygb*^{801a} (Fig. 3L).

The part above was added to the discussion in lines 317-326.

6. The expression of CYGB in epithelial cells is not generally accepted as almost publications up to date showing CYGB expression in mesenchymal cells, splanchnic fibroblasts, pericytes. In the lung and heart, its expression in stromal cells (Lab Invest. 2004 Jan;84(1):91-101; Clin Mol Hepatol. 2020 Jul;26(3):280-293). This study used two antibodies anti-CYGB to detect CYGB expression in the mice, however, one of them, are specific for only human tissue (Sigma HPA017757).

We have clarified these issues in the revised manuscript. We agree that the epithelial cells do not show high levels of CYGB compared to mesenchymal cells, however we do detect CYGB expression in epithelial cells.

The immunostaining shown in previous Fig. 5A was performed using the rabbit polyclonal anti-cygb antibody (Sigma HPA017757) that was validated previously in the Human Atlas. These data show detectable levels of CYGB expression in human samples by single cell RNAseq in airway ciliated cells and positive immunostaining in broncheal airway epithelium. The data are available in the human atlas website here:

<https://www.proteinatlas.org/ENSG00000161544-CYGB/single+cell+type/bronchus>
<https://www.proteinatlas.org/ENSG00000161544-CYGB/tissue/bronchus#img>

In order to test the affinity of the sigma antibody for mouse *Cygb* we analyzed the trachea lysate from wt (*Cygb*^{+/+}), heterozygous (*Cygb*^{+/-}), and homozygous (*Cygb*^{-/-}) mouse mutant by western blot. Our results show a detectable 21 kDa band corresponding to *Cygb* in wt and heterozygous mice, and lacking in the KO lysate. We added in the manuscript the WB data and the immunohistochemistry data performed with the same antibody in supplementary Fig. S8B-C. Additionally we added in Fig. 5A a better representation of *Cygb* detection in airway cells performed by immunofluorescence which supports our data that *Cygb* transcript is detectable in airway cells by RNAscope (Fig. S7 E-F).

Another important evidence that CYGB transcript is detectable in ciliated cells in humans is made available also in the IPF Cell Atlas at:
<http://www.ipfcellatlas.com/>

We also have provided additional new data on human airway cells obtained from scraping tracheas collected from two healthy humans. We could detect a 21 kDa corresponding to the human CYGB monomer, as shown by comparison with the recombinant CYGB (supplementary Fig. S 7A).

Reviewer #3 (Remarks to the Author):

We appreciate the important constructive suggestions from reviewer 3 and have responded to all with new experimental data as well as methodological and editorial clarifications.

The manuscript by Rochon et al seeks to address an important mystery in LR development, primary ciliary dyskinesia, and NO signaling. For years, clinicians have measured NO levels from the nasally exhaled air of patients and noted that patients with PCD have low levels of exhaled NO compared to healthy patients. This is an important clinical test as patients with PCD do not always have other (more obvious and easily identified) symptoms of PCD such as situs inversus or heterotaxy at presentation. However, they may suffer from respiratory illness or infertility. Given the importance of diagnosing poor respiratory clearance in these patients and providing mucus clearance support/aggressive pulmonary infection treatment, rapid testing via NO has proven highly useful. Given that we simply have no idea how PCD relates to NO signaling, the potential impact of this manuscript is very high. Therefore I am generally enthusiastic of the discovery outlined in this manuscript.

We thank the reviewer for these supportive comments. We also agree that the additional mechanistic data indicated by the reviewer can provide very important insights into NO signaling, cilia structure, and PCD. We have performed extensive additional experiments as well as clarified future areas where more work is required. *Our primary findings that a globin can positively regulate NO synthesis and cilia function is completely novel and has the potential to open a new field of investigation.* Recent discovery of a new globin called androglobin in motile sperm and extensive human atlas observations of cytoglobin expression in epithelial cells already provides compelling suggestions that heme proteins can mediate important processes for cilia function and our findings will be very relevant. Our new experimental data are summarized with each suggestion by the reviewer below:

Currently, the manuscript seeks to understand the role of cytoglobin in embryonic development. This is clearly an unexpected and exciting discovery that would be of interest to the field of NO signaling but not to the broader audience that Nature Comm serves. Previous work has demonstrated that cytoglobin may play a role in NO homeostasis. cygb2801 mutant embryos have defects in LR. The authors then do a nice job phenotyping and demonstrating that other NO pathway members also give a similar phenotype. However, what is lacking is a mechanism that connects NO signaling to a defect in cilia structure or precisely how NO leads to a lack of motility. I could see a few mechanisms – alterations in notch signaling that lead to specification of immotile cilia vs. motile cilia, alteration in cilia assembly that lead to loss of the inner double of microtubules, there are certainly others as well. Currently this is just a phenotype associated with the mutant embryos. Unfortunately, without some insight into the mechanism directly connecting NO signaling and cilia function the paper falls short of the bar for Nature Comm.

Our initial findings support the involvement of the NO-related signaling pathway on cilia function, although more data may be helpful to unequivocally support this claim. To address this issue we present unexpected mechanistic data that cytoglobin positively regulates the Nos2b-NO-sGC-cGMP axis and that this signaling axis regulates development, ciliary function and laterality determination. This in itself provides extensive mechanistic data. We have performed additional work described below to add more mechanism that our expert reviewer has requested. We hope this is sufficient considering the novelty and robustness of our findings.

Concerns:

1. The paper seems to begin with cytoglobin as the discovery. This reduces the impact of the paper – really this should be about NO signaling and LR patterning. Cytoglobin is an exciting avenue that leads to this but really the impact of the paper is on NO signaling and mechanisms of PCD.

We respectfully challenge this assertion. A major mystery in this field of science is why globins are expressed in many cells outside of red cells and muscle cells. Our primary work focused on understanding the role of cytoglobin in biology, which was the starting point leading to this discovery and has been an unsolved mystery for more than 20 years since CYGB discovery. A role in development, positive modulation of the NO axis, and a role in regulating cilia was completely unexpected and will create new fields of inquiry.

Despite the evidence that normal cilia motility is related to NO levels, the mechanisms of NO production – possibly relevant NOS isoform(s) – remains unknown. Our extensive data reveal that *Cygb2* co-localizes with cilia and the NO synthase *Nos2b* in the Kupffer's vesicle, the fish laterality organ, and the structure and function of cilia were disrupted with *cygb2* knock-out, abolishing fluid flow within the Kupffer's vesicle. Abnormal ciliary function and organ laterality is phenocopied by depletion of *nos2b* and *gucyl1a*, one of the NOS isoforms and the canonical NO receptor soluble guanylate cyclase (sGC) homologs in fish respectively, and rescued by exposing *cygb2* knock-out embryos to an NO donor, an sGC stimulator and with over-expression of *nos2b*. Consistent with a conserved role in regulating cilia function, *Cygb* knock-out also impaired mouse airway epithelial ciliary structure. We also found cGMP to be involved in the *Cygb*-NOS-NO pathway to determine laterality. Thus, *Cygb* appears essential for normal development stage-specific NO signaling, ciliogenesis and ciliary motility, required for the establishment of left-right patterning. These studies identify cytoglobin and the NOS-NO-sGC-cGMP pathway, as shown in figure 4, as an up positive regulator of cilia, opening the door to new therapeutic applications where NO donors and sGC activating drugs can enhance ciliary function.

2. There is a brief description that somites and craniofacial development are abnormal in *cygb2801* mutant embryos. Does this phenotype help identify potential mechanisms? The craniofacial cartilages are products of an interaction with the neural crest and the surrounding mesenchyme and requires a series of interactions with different signaling processes (Wnt, BMP, etc). Could this be an avenue for mechanism discovery that could also be applied for LR patterning? I am not suggesting that the authors expand work on craniofacial or somite patterning unless it informs their work on LR. In fact, I would suggest that they remove craniofacial patterning and somite analysis unless they can link it to the underlying signaling mechanism – I believe the authors are trying to be thorough (which is laudible), but the manuscript is focused on LR patterning which is enough.

As requested, in the attempt to thoroughly characterize the zebrafish phenotype, we initially described the craniofacial defects in the *Cygb2* mutants. However, we agree that the focus of this manuscript should be on left-right patterning related to cilia function. This also aligns with reviewer 1 concerns and suggestions. We have removed the craniofacial defects data from the manuscript for further detailed study and have focused our results in a more conservative fashion on left-right patterning.

3. There is also a comment on shortened body axis. Looking at the fish it appears that they may have scoliosis(?). Scoliosis has been associated with cilia dysfunction in fish and if this is the case might be worth discussing especially if it is informative to the LR story.

We thank the reviewer for pointing this out! This observation is indeed correct. Following the reviewer's suggestion, we analyzed the curvature of the vertebrate spine in the *Cygb2* mutant adult zebrafish and found a significant number of individuals with spinal curvature compared to wt, further supporting a role for *Cygb2* in cilia function. These new data are added in Fig S1F.

4. Fig 2 – the authors state that *Cygb2* is expressed in the cilia. Frankly the relatively low mag imaging is not entirely clear. A high magnification would help a lot in convincing the reader. Additionally, the imaging should be done in the *cygb2801* mutants to demonstrate the signal is lost and therefore specific. There is well known false positive where primary antibodies mixed with acetylated tubulin primary antibody lead to cilia signal that is false. I just realized that they have immunolocalization in figure S3a in the context of control MO and *cygb2* MO injected embryos. Looks like the signal is reduced but hard to tell at this low magnification. This could help. Is the antibody used for western blot the same as the immunolocalization?

In response to this important concern, we added in supplementary Fig. S2A and S3A a higher magnification of KV cilia observed in the *cygb2⁸⁰¹* mutant and *cygb2* morphant respectively, co-stained with two different anti-*Cygb2* antibodies and anti-acetylated tubulin. *Cygb2* staining is undetectable in mutants and morphants despite acetylated tubulin staining is still present. To detect *Cygb2* we used two custom antibodies, one raised in guinea pig generated from the recombinant protein (Fig. S3A) and one synthesized against peptides 113-130 made in rabbit (Fig. S2A).

5. More importantly, if cilia localization of *cygb2* is true, what does it mean? Does this mean that cytoglobin is affecting NO signaling in the cilium itself? How does that affect cilia structure given that the cilia appear smaller? The authors do not discuss intraciliary NO signaling and what this might mean? This is critical for understanding mechanism.

The reviewer makes an important point. Cytoglobin, via the positive modulation of NO, appears to affect *i*) cilia function as well as *ii*) cilia structure. In our revised document we recognize this and focus on the NO signaling effects on both properties. We also point out that NO has been known to modulate ciliary beat frequency (Jain et al. *Biochem Biophys Res Commun.* 1993 Feb 26;191(1):83-8). However, the mechanisms underlying NO signal transduction in cilia are unknown. NO signaling can occur via numerous possible pathways including post-translational modifications, reactive oxygen/nitrogen species production or through sGC activation. In the present manuscript we provide strong evidence that the mechanism involved is through sGC-cGMP. A number of downstream pathways may be affected by altered levels of cGMP such as Hedgehog signaling (Christensen et al. *Stem Cells Dev.* 2006 Oct;15(5):647-54), Wnt (Ma and Wang. *J Biol Chem.* 2006 Oct 13;281(41):30990-1001) or the

intraflagellar transport machinery (IFT) assembly (Muthaiyan Shanmugam et al. Mol Cell Biol. 2018 Mar 15;38(7):e00612-17) that could certainly lead to alterations in ciliogenesis, as suggested by this reviewer. These studies are in progress in our lab and potential new discoveries will require extensive validation to be developed in a future manuscript. For this first manuscript we suggest a primary focus on cGMP effects on function (cilia function, fluid flow in laterality organ, cilia structure).

6. Fig 2 – Amazingly, *cygb2* appears to be localized to the right of the LRO (left right organizer) (I think – the images in 2b lack orientation of what is anterior-posterior/left-right – this needs to be corrected).. Is *cygb2* expressed symmetrically at earlier stages? For example *dand5* is expressed symmetrically at earlier stages and then becomes asymmetric at later stages (*dand5* becomes right sided with reduced expression on the left due to LRO flow). Is this also true for *cygb2*? This would be utterly fascinating and would suggest that NO signaling is also lateralized. If so this would be downstream of cilia signaling. This needs to be addressed.

This important observation was also pointed out by reviewer 1. We have now thoroughly analyzed the staining in the KV by RNAscope and did not detect any significant differences in *Cygb2* left-right expression across all experiments. Orientation labeling for left-right/anterior-posterior was added to the figure.

7. Examination of the expression of *cygb2* is essential at earlier stages. Given the asymmetric expression of *cyb2* at 10 somites how do the authors explain the cilia expression? As above, is the expression symmetric across the LRO or not at earlier stages that then become asymmetric? Is it responsive to LRO flow or not – please take a look at martin blum’s paper in *Xenopus* about *dand5* (aka *coco*). Also the recent paper by Yuan in *Science* also takes advantage of *dand5* in zebrafish. Also looking at *nos2b* – it would be interesting to examine this gene’s mRNA expression at earlier stages as well.

We agree that monitoring *nos2b* expression throughout the KV development would determine whether any asymmetry in NO signaling is responsible for the left-right pattern. If so, NO detection would also be asymmetric. However, with further analysis of all embryos we do not see an asymmetric distribution of cytoglobin in the KV. This evidence suggest that NOS/NO may have similar symmetric expression.

8. The LRO has two types of cilia - motile and immotile cilia. As the authors suggest manipulation of NO signaling appears to shorten the cilia – this could represent a transformation of the cilia from a motile cilium into an immotile cilium. This has previously been implicated by Notch signaling (see the papers by Lopes lab and Khokha lab). Therefore, one explanation for the change of length could be a change in notch signaling and a change in cilia type. Another possibility is that as the authors seem to suggest there is a change in cilia structure. It is a bit tough to tell what is going on with the quality of the TEM – perhaps the loss of the central pair is a change from cilia type (ie transformation to an immotile cilium). Unfortunately the TEM is not of very high quality. Certainly getting good TEM of the LRO is technically challenging. More importantly how does one connect a reduction in NO signaling to this change in cilia structure?

We think that this is an intriguing possibility, particularly because Notch has been shown to upregulate *Cygb* expression in the vasculature (Lilly et al., *Vascular Pharmacology*, 2018 110,7-15). We plan to further study this possible link in future studies although is beyond the scope of the current work.

9. In Fig 2, the authors nicely test that mutants of *cygb2* have reduced levels of NO and that this can be rescued by the expression of *cygb2* mRNA. This leads to a number of questions: 1) does the gain of NO by GOF of *cygb2* affect LR development or development in general – this might give clues to its function 2) does other models of PCD cause decreased levels of NO – for example, *dnah9* depletion or *c21ORF59* depletion? What about loss of cilia? This is relevant since patients with PCD also have low levels of NO and the authors are supremely positioned to test if they can make zebrafish a model for testing levels of NO and understanding mechanism. This could create a whole new avenue and answer a critical question in the field – why do PCD patients have lower NO levels? Additionally, they could test if adding NO agonists in that context, rescues the PCD phenotype! This might be exceptionally impactful.

We agree with the reviewers comments, and especially the therapeutic potential of upregulating NO signaling with NO donors or sGC activating drugs that are currently FDA approved. We add a discussion of this and point to our data using the NO donor DetaNO to enhance cilia activity in the Kupffer's vesicle and restore laterality. This may also be very applicable to hypomorphic variants of human PCD, like *RSPH1* (Yin et al., *Am J Respir Cell Mol Biol* (2019)), *DNAH11* (Knowles et al., *Thorax* (2012)) and *CFAP221* (Bustamante-Marin *J Hum Genet* (2020)). Mutations in these genes are reported to result in a "hypomorphic or mild" PCD phenotype and none of them exhibit apparent cilia defects by TEM analysis. Of note, *DNAH11* mutations are commonly detectable in 6-9% of the PCD cases (Shapiro et al., *Am J Respir Cell Mol Biol* (2022)).

A whole body of research for many years have been focused on understanding mechanisms of NO production and regulation, but this still remains a mystery particularly in airway cells. Other globins (although not *Cygb*) have been found to interact with NOS to regulate NO release (Straub et al., *Nature* (2012) 15;491(7424):473-7); Lechauve et al., *JCI* (2018) 128(11):4755-4757). Our study presents cytoglobin as unexpected player involved in NO production, interacting with the NO synthase in cilia. In this regard we performed new experiments to verify the activity of NOS in presence of *Cygb*. Our new data shown in Fig. 5F suggest that oxygen bound to *CYGB* in combination with iNOS favors the production of L-citrulline from L-arginine to produce NO. This unexpected finding lay the ground for a whole new investigation on *CYGB*-NOS interactions that could be targeted for therapeutic applications in PCD. Testing whether modulation of this upstream pathway can rescue amounts of NO levels in PCD patients is certainly an attractive strategy that we will explore in our future work.

Our new findings and discussion are added to the manuscript in lines: 254-269; 320-323; 324-326.

10. Fig 4E – this is confusing. The authors state that DETA/NO treatment completely rescued laterality defects in mutants at a concentration of 250 μ M – effect is dose and stage specific when embryos are treated prior to gastrulation – however according to the time chart the embryos treated at 2-4 cells and oblong are before gastrulation and have cardiac looping defects similar to untreated mutants while those treated later are rescued – dome and epiboly? This doesn't make any sense. No statistical testing is done.

We realized that this was confusing so we rephrased the description of the data in text (lines 220-221) to clarify: the rescue effect of DETA is detectable when starting the treatments at the beginning of gastrulation (dome) or 50% epiboly, prior to the progression of it, while treatments at earlier stages during blastulation have no effect on cardiac looping. Statistical analysis by Chi-squared test has been added in all the figures where previously missing.

11. Mouse work – Fig 5A – this is not helpful. We need a higher magnification image of the A to be able to see the cilia.

We agree that a higher magnification would be more informative to the purpose of cilia visualization. We added the immunofluorescence staining for *Cygb* and Acetylated Tubulin in new Fig. 5A that should clarify this point. We moved the immunostaining and visualization with DAB in suppl Fig. S8B and we added the WB analysis of mouse trachea lysate (Fig. S8C) as this helps the response to reviewer 2 in regards to the antibody used.

Minor Concerns:

1. Two alleles generated from the same set of two guide RNAs were used to create the mutants used in this study. Ideally, the two alleles would be generated from guide RNAs that do not overlap. Unlike random mutagenesis, CRISPR may hit the same off target so rather than make two alleles from the same sgRNAs it would be preferable to make a second allele with non-overlapping sgRNAs. To be clear, reduction of protein expression demonstrates that the LOF alleles are effective but not specific – there may still be a second site that is mutated that is causing the phenotype.

We generated the two mutant alleles from two gRNAs that although overlapping, they only share 9 bases and they were designed to target different PAM sequences. We agree that the possibility of off target sites is a concern but we think that we minimized this possibility by outcrossing the founders with wt and by selecting the mutants together with their wt siblings for controls. These animals were bred for at least 3 generations before being used for experiments and the same phenotype is observed in both mutant lines. In addition to this, injections of *cygb2* morpholino fully phenocopies the cilia and laterality defects observed in the knock-out, further supporting the specificity of this phenotype.

2. Western blot – Fig S1 – better annotation is needed. Clearly the antibody detects a couple of bands in fish – I believe the band that the authors wish to highlight which is missing in *cygb2801a* is running a bit higher than the *Cygb2* recombinant protein. Of note this band remains present in the *cygb2801b* allele which should predict a much more severe truncation? What is going on here?

We apologize for the confusion and have further clarified this point in the revised manuscript. Our recombinant globins have a molecular weight about 3 kDa higher than endogenous ones due to the His-tag that they all carry for purification purposes. As this may be confusing we removed the recombinant protein and we repeated the experiment to have a better quantification of the band corresponding to *Cygb2* in *cygb2^{801a}* mutants. These results are added in Fig. S1B.

3. SFig 4 - no statistics tested in the graphs. This needs to be corrected. Is there synergy between *Nos2b* and *cygb2*? In SFig4a – looks like the incidence of the cardiac loop is lower in the *cygb2801a* allele when *nos2b* MO is applied – does this make sense?

Thank you, we have added statistical tests as requested. We observed no statistically significant difference between the two categories in supplementary figure S4A.

4. SFig5 – again no statistics in any of the graphs.

We added statistical data in the graphs as noted.

5. The heart is normally on the left. But the cardiac loop goes to the right (ie dextral looping) this is the standard nomenclature in the field. Left heart loop would be the reverse but it seems the authors are not using this in the conventional way – Fig 4F,G for example – the heart should be on the left and the cardiac loop should be rightward. This is pretty basic in the field – might be of some use to get a LR expert to review some of these data to make sure the nomenclature is right – otherwise this is pretty confusing.

We thank the reviewer for pointing this out. We corrected the graphs to “left heart” on the Y axes instead of “left heart looping”.

6. A number of figures could benefit from additional labeling. Many times it is difficult to follow the figures without reading the figure legends when just some simple labels to the figures would improve readability considerably.

To address this important issue we have added the orientation (left-right, anterior-posterior) to the figures illustrating the KV.

Reviewer #4 (Remarks to the Author):

We appreciate the important constructive suggestions from reviewer 4 and have responded to all with new experimental data as well as methodological and editorial clarifications.

This is a novel manuscript that reports that Cytoglobin 2 (Cygb2) in Zebra fish regulates ciliary motility and organ laterality. This is new, important, and unexpected as the authors note. It is clearly shown that genetic deletion of zebra fish Cygb2 contributes to cardiac and GI tract laterality defects. In zebra fish Cygb2 colocalizes with NOS2b and modulates NO sGC signaling with this affecting ciliogenesis and cilia motility that is required for left-right patterning in zebra fish. This effect on modulating embryogenesis is new and very interesting. The manuscript presents a large amount of data and attempts to span implications from Zebrafish to mice to man. These efforts to seemly over extend the implications of the work is a weakness of an otherwise largely solid and strong paper.

We thank the reviewer for positive comments on the importance of the work and have focused our interpretation and studies more on development and the laterality findings in zebrafish. Thus we removed some of the mouse airway epithelium data including the cilia beat frequency as this will be developed in our future studies using the mouse KO to determine the role of cytoglobin in Kartagener’s syndrome.

While Cygb deficiency in Zebra fish is compared to Kartagener’s Syndrome in humans, this seems valid in some ways including situs inversus and ciliary dysfunction but perhaps not in others such as sinusitis, bronchiectasis. In mice comparison is made to Cygb^{-/-} mice where ciliary dysfunction is seen but these mice do not have situs inversus but were found to have ciliary dysfunction in the current study. One can not assume or expect that the properties seen in zebra fish precisely extend to mice and man. The properties of Cygb2 in zebra fish may be somewhat different than Cygb in mammalian systems. In fact there are two Cygb isoforms in zebra fish and here only Cygb2 is studied as it has 75% homology to mammalian Cygb, while Cygb1 has 63% homology. Only Cygb2 is studied here with CRISPR/Cas9 knockout; however, the role and properties of Cygb1 are not studied or commented on.

It is stated several times that the mechanisms underlying Cygb function are unknown (as in line 77) and that its physiological function is unknown (lines 33, 241). This is inconsistent with the prior work of this group and others in the field. Indeed the paper properly presents the functions identified to date, as a NO dioxygenase, nitrite reductase, redox sensor, and antioxidant / SOD function. The role of Cygb in embryogenesis, development, and ciliary function is new and important; however, clearly there is much prior data on the role and functions of Cygb. As such the authors would be best to reword these lines to focus on what is new here without detracting from what is already known and established.

We acknowledge that prior work exists that provides important insights into globins function. A large majority of these studies (including ours) are limited to mechanistic data provided by in vitro analysis. We agree with the reviewer that in the text we should focus on what is new here and its role in development. We rephrased the lines in the text referring to prior data and acknowledged the mechanistic studies on Cygb. These changes can be found on line 35-36, 73, and 271-272

We appreciate the comments regarding Cygb1. In response to this important concern we have included new data on our Cygb1 mutant that can be found on lines 107-112. Further details are described in the answer to question 2. In addition, we have characterized Cygb1 and Cygb2 at length and we demonstrated that the properties of Cygb1 and Cygb2 are very different, and Cygb2 is by far more similar to the mammalian protein (Corti et al., Nitric Oxide (2016) 53:22-34; Amdahl et al., Biochemistry (2019) 58(29):3212-3223). Cygb1 appears to be 5-coordinate with higher oxygen binding affinity compared to the 6-coordinate Cygb2.

There are several specific concerns that should be addressed by the authors as listed below.

1. Line 33 “unknown physiological function“ this should be reworded, such as “questions remain regarding its functions. Its role in development and embryogenesis is unknown”

We have corrected the text as indicated on lines 35-36.

2. The paper focuses on Cygb2. Was Cygb1 also measured? Does it also localize with cilia? Where does it localize?

We did focus on Cygb2 as this is the gene that seems more similar to the mammalian protein and thus may provide more clues about the conserved roles of cygb through evolution (Corti et al., Nitric Oxide (2016) 53:22-34). However, we have studied the expression of Cygb1 as well, and we include some of these results in the supplementray material.

We and others showed that *cygb1* and *cygb2* expression is spatially and temporally different in zebrafish embryos (Corti et al., Nitric Oxide (2016) 53:22-34; Tiedke et al., J Comp Physiol (2011). In these studies Cygb1 was found in larvae older than 24 hours post fertilization and surprisingly in adult red blood cells, consistent with our observations that Cygb1 displays all the characteristics of a respiratory globin, perhaps representing the evolutionary transition between 6-coordinate and 5-coordinate globins.

To address any possible physiological interference of Cygb1 with Cygb2, using CRISPR we successfully generated the *cygb1* mutants (*cygb1*⁸⁰²) as shown by the lack of protein in the new western blot (Fig. S1B).

We screened for the mutation (8 bases deletion) in the F0, selected founders and bred them for three generations before using them for these experiments. We detected normal viability in F3 maternal zygotic *cygb1*⁸⁰² knockout by Kaplan Meier analysis (Fig. S1H). To detect possible redundancies or compensation between *cygb2* and *cygb1* we 1) observed no change in Cygb1 protein expression levels in *cygb2* mutants and 2) analyzed the left-right pattern in *cygb1*⁸⁰² mutants and found no abnormalities in cardiac orientation. This excludes the possibility that Cygb1 function overlaps with Cygb2 function. We added this analysis in Fig. S1B and I.

3. On lines 69-71 Again, it is suggested to simply say that questions remain regarding its precise range of functions and its role in development has not been studied.

We have corrected this section as indicated on line 73.

4. On line 79 here or in the discussion, it would be good to comment on similarities and differences in structure compared to human Cygb particularly in the vicinity of the heme.

We have added information on line 82-83 regarding the heme coordination differences between Cygb1 and Cygb2 and similarities between Cygb2 and mammalian Cygb. The structures of zebrafish Cygb1 and Cygb2 have not been yet determined, so any structural discussion is speculative. AlphaFold models are available, however these models lack the heme moiety. Given the high sequence identity and similarity between Cygb1/2 and mammalian Cygbs, it is hard to predict a priori significant differences in the heme pocket; the main heme ligands Phe B10, His E7, Val E11, His F8 (standard globin nomenclature) are identical in fish or mammalian Cygbs. Other residues important for the binding of the heme carboxylate groups, like R84 or K116 (human Cygb numbering, Kaliszuk et al. Nitric Oxide (2022)) are also conserved, although K116 is replaced by an Arginine in Cygb1. Altogether, it is difficult to predict substantial differences based on the existing sequence data. In fact, the weak coordination of the distal histidine that we previously observed in Cygb1 was unexpected (Corti et al.; Nitric Oxide (2016) 53:22-34). For this reason, we have declined to comment in the paper on structural comparisons and hope the reviewer understands.

5. Line 88, with regard to the organ laterality why is this only 87.3% in control? Is this expected? In the discussion, it would be good to discuss what a given % change implies, that if Cygb2 was essential for laterality this would decrease to 0, so it seems to contribute to this process of development but is not essential for it.

This is an important piece of evidence that is recurrent in many studies centered on laterality development in zebrafish. It is not clear the reason why ~ 10% of normal embryos intrinsically develop laterality defects but it is expected as per previous literature (Lahvic et al., Dev Biol (2013); Wu et al., Dev Biol (2021); Saydmohammed et al., Nat Comm (2018)). We have noted this in the text on line 95.

6. Line 88-89, this data as stated does not seem to be consistent with supplement figure 1D, please check this and explain this.

We observed variability among experiments and the percentages of left heart may fluctuate. To determine whether the differences between wt and mutants were significant, we averaged the data from each experiment (typically we pool ~ 50 embryos to reflect the experimental set up) and we plot averages from

different experiments. This representation is illustrated in Fig 1C-D and described in lines 92-93 as percentage with left heart. In new supplementary fig S1C we present the percentage of embryos detailing left, right or bilateral asymmetry based on the number of individual embryos analyzed, which is reported in red on top of the bars.

7. Lines 135-145 and figure 2 F & G, there are several concerns here and a need for additional information. First of all, NO can be oxidized to nitrite or nitrate, with nitrate usually the major product due to NO dioxygenation by heme proteins such as Cygb. Since Cygb is the focus of the paper, it seems odd to ignore this. As such, it would be logical to measure both. Here only nitrite is measured. Alternatively, one could directly measure NO.

We focused on nitrite as a surrogate of NO synthesis as nitrite is the primary oxidation of NO whereas nitrate may be generated by NO dioxygenation but can also derive from many other sources, including the water and dietary sources. The measure of direct NO in tissues is challenging due to its instability and high interference by several biological components. It is particularly difficult in zebrafish embryos because of the extremely limited amount of tissue. The only methods utilized in the developing zebrafish are chemiluminescent methods by diaminofluoresceins such as DAF FM in whole mount embryos (Lepiller et al., FRBM (2007)). Even if this could give an indication of NO localization, one cannot rely on these methods to directly quantify differences in NO concentrations. The reactivity of diaminofluoresceins is not fully understood, secondary reactions with ascorbate and thiols may occur in cells resulting in non-specific signals and sometimes high background. The spontaneous oxidation of NO gives nitrite, while the enzymatic deoxygenation of NO gives nitrate. We opted for the NOA to quantify nitrite/NO as best and we found lower nitrite in cygb2 mutants compared to wt. Our data does not support a major NO dioxygenase role for Cygb2, since in that case NO would be converted into nitrate and, we would detect lower NO (as nitrate) in wt compared to mutants. Instead, we measured the opposite. In the present study, we could rescue the laterality phenotype by treating the embryos with NO and by injecting a Nos2b mRNA construct into the Cygb2 mutants. This support the hypothesis that Cygb2 plays a role in favoring NO production, rather than scavenging, specifically through possible interactions with Nos2b. As suggested by other reviewers we added more mechanistic data on this. We performed new experiments to verify the activity of NOS in presence of Cygb. We explored CYGB reduction by Nos2b and mammalian iNOS in the presence of NADPH, and found that CYGB was efficiently reduced by iNOS/NADPH, suggesting formation of transient complexes between CYGB and iNOS (Fig. 5D-F). This supports a mechanistic interaction between these proteins. In addition, we analyzed iNOS activity in presence of CYGB and found that CYGB increases NO synthesis by iNOS at lower oxygen tensions (Fig. 5F). This unexpected finding lays the ground for a whole new investigation on CYGB-NOS interactions. Excluding a role for Cygb2 in NO dioxygenation, the amount of nitrite present in the system should be an indication of the NO present at the time of harvest.

There are also technical questions and concerns regarding the nitrite/NO measurements of Figure 2. In Fig 2 G the units are shown as nM/ug, it seems that ug is the protein concentration but this is not defined here. The protein concentration is specified on the x-axis as 50 or 100 pg. It is unclear what "co" is as this is not defined. Please define what "co" is. With 50 pg or 100 pg of protein with the measured nitrite of 0.1 to 0.3 nM/ug protein, the levels would be well below the sensitivity detection limit of this technique. Please check this figure and explain this. In general the detection sensitivity limit of this method even with larger volumes is about 10 nM at best. Another related problem is that typical buffers can contain 0.1 to 1 uM amounts of nitrite and this background level would well exceed the levels noted. Again please clarify

the conditions here including the protein concentration, the volumes used and background nitrite levels. Also for the WT, the observation number of 3 or 4 points for 50 and 100 pg seems insufficient in view of the variability.

We appreciate the opportunity to clarify this. In Fig. 2G the Y axes represent nitrite concentration normalized per amount of protein injected, thus nM/ug of protein. The X axes shows the different groups of embryos injected with cygb2-mRNA in the amounts of 0-50-100 pg, where 0 is the control (uninjected). We apologize for the confusion and we changed the figure and legend to clarify.

We typically inject in the NOA ~ 800 ug of protein lysate, (from a pool of 50 embryos) that corresponds to 0.2 uM nitrite. We normalized this to the amount of protein injected to obtain about 0.2 nM nitrite per ug of protein. Since our samples are characterized by very limited amount of nitrite/NO, we are aware of the possibility that the buffer may contain more nitrite than the samples preventing from detecting small differences in concentrations. To avoid this we make fresh nitrite and we use all the experimental cautions to avoid nitrite contamination. The nitrite in our buffer is typically less than 0.1 μ M and this background is indeed taken into account when doing calculations.

8. Line 169-170 and lines 179-180, while Nos2b knock-down phenocopies the Cygb mutant, this does not prove a direct Cygb-Nos interaction. As noted in the paper and shown by this group and others, Cygb can be a potent nitrite reductase under hypoxic conditions, as Nos is a major source of nitrite, it could be required for NO generation by Cygb2. That is there could be 2 parallel pathways of NO generation with Cygb requiring NOS as a source of nitrite.

This is possible but seems a bit counterintuitive as NOS would be making NO that would get oxidized to nitrite and then reduced back to NO with no net gain of NO. Our new data with the L-citrulline assay (please see new Fig 5F as discussed above) demonstrates the ability of oxygenated CYGB to induce iNOS-NO production. This is another piece of evidence in support of CYGB-iNOS interactions.

9. What is the oxygen level in these zebra fish embryos? Measurement of this would be helpful to understand how Cygb2 is functioning here.

We agree that this would be of great help to understand Cygb2 function in zebrafish embryos. Although there may be some colorimetric assays available, we are not aware of any specific method that could allow a precise measurement of oxygen concentration in such small samples.

10. Line 185 the spelling error “suggestiong” should be corrected to “suggesting”

We changed this as indicated.

11. Supplemental figure 5 B, 0 nM SOD should be 0 Cygb2.

Thanks for noting this, we have corrected this error.

Interestingly, it appears in 5C that SOD+catalase decreased laterality. Is this so?

Mammalian *Cygb* has been shown to act as a superoxide dismutase in SMC (Zweier et al., PNAS (2021)). To verify if this was the case also for the zebrafish *Cygb2* we analyzed SOD activity by *Cygb2*, *Cygb1* and human CYGB and found none (supplementary Fig. S5B, E-F). Additionally we treated the embryos with SOD+catalase or SOD mimetic to possibly restore the correct cardiac laterality and we found no significant differences between *cygb2*⁸⁰¹ and *cygb2*⁸⁰¹-treated. Statistical analysis was added to the supplementary Figures S5C-D.

12. Discussion, Line 241 Again suggest to change this wording

We rephrased this as suggested.

13. Where Kartagener's syndrome is noted, it would be good to note similarities and differences and what is known about the molecular basis of this disorder. Do the authors intend to suggest that there is a defect in *Cygb* expression or structure in these patients?

Our data suggests that *Cygb* is an upstream positive regulator of cilia function through activation of NOS-NO-sGC-cGMP. This is very novel as a globin has not been shown to positively regulate NOS or cilia. Based on this, it is possible that a primary mutation in cilia genes may be associated with compensatory upregulation of the CYGB-NOS-NO-sGC pathway. Testing some of the PCD genetic variants, particularly the ones responsible for "hypomorphic" phenotypes, is an attractive strategy that would allow to determine the impact of these variants on CYGB-NOS-NO-sGC signaling pathway and to assess whether upstream therapeutic activation of cytoglobin, NOS, NO, or sGC can improve cilia function in these human PCD "hypomorphic" mutations. These studies will be undertaken in our lab and will be developed in future work.

14. Line 247 In view of the prior work from the authors and others in the field on the potent NO dioxygenase function of *Cygb*, how do the authors explain the lower levels of NO and nitrite reported. As in the presence of oxygen *Cygb* will efficiently scavenge NO and likely nitrite will be depleted as well, how can this be explained? Is it proposed that the embryos are markedly hypoxic? If so *Cygb* would be expected to function as a nitrite reductase?

Likely the level of oxygen in fish embryos is lower than terrestrial animals. The ability of *Cygb2* to produce NO from nitrite is certainly a possibility but our new data on CYGB reduction by iNOS in the presence of NADPH and the new L-citrulline assay of iNOS activity in presence of CYGB suggests that *Cygb* plays a role in NO production through NOS regulation rather than through nitrite reduction. We agree with the reviewer that the results are unexpected given the known in vitro capabilities of *Cygb*. The lower NO levels in the *Cygb* KO suggest that *Cygb* is involved in NO generation, so we also suspected a nitrite reduction mechanism. However this is not consistent with our results. Our new data suggests that given the low oxygen tension in the tissue, *Cygb* may assist NOS function by providing oxygen for NOS-dependent NO synthesis

15. Line 293 regarding *Cygb* modulating NO production in cilia through a NOS-dependent process, so far there is no established basis for a direct *Cygb* NOS2 interaction. It is more likely that this is due to NOS2 being a major source of nitrite, but the NO production would require a low oxygen tension. Perhaps this exists in the zebra fish embryos. It would be interesting to see if one adds nitrite with the *nos2b* knock down if this can restore NO generation and laterality.

To add more mechanistic data, we performed spectrophotometric evaluation of Cygb reduction by NOS in the presence of NADPH, and found that Cygb was efficiently reduced by iNOS/NADPH and Nos2b/NADPH, suggesting formation of transient complexes between Cygb and iNOS (new Fig. 5D-E), with associated electron transfer reactions. This is the first evidence in support of a mechanistic molecular interaction between Cygb and NOS. In addition, we analyzed iNOS activity in presence of Cygb and found that Cygb increases NO synthesis by iNOS at lower oxygen tensions, possibly by providing oxygen to NOS for de novo NO synthesis (new Fig. 5F). Altogether our data favor the Cygb-NOS as main source of NO.

16. Lines 305 and 320 It is not clear that Cygb is modulating NOS function, it seems more likely that NOS is modulating Cygb-mediated NO production. All of these processes both Cygb function and NOS function are oxygen-dependent, thus there is a critical need to measure the oxygen levels in this system to understand the underlying regulation of NO generation.

We agree that oxygen levels can be critical to shift Cygb from a dioxygenase role to a nitrite reductase role. To address this point, we performed an experiment (Fig. R3) where we treated the *cygb2⁸⁰¹* with nitrite to possibly restore the normal cardiac laterality however, no significant effect by nitrite was observed. Oxygen levels in the tissue are hard to determine accurately. In addition, Cygb2 has a high O₂ affinity (p50 4.4 torr at 25 C per (Amdahl et al., Biochemistry (2019) 58(29):3212-3223) and will be at least partly oxygen bound even at low oxygen tensions. Given the relatively low oxygen affinity of NOS (Santolini et al., JBC (2001) Dec 28;276(52):48887-98) we find more probable that Cygb is acting as an oxygen supplier for NOS-dependent NO synthesis as discussed above.

17. In spite of the current limitations in mechanistic understanding noted above, the fundamental effects of Cygb2b knockout on embryogenesis are of high importance and novelty in themselves. Thus, addressing and considering the above comments may be sufficient for the current manuscript as there are limits as to what amount of data can be expected in one manuscript and the paper already seems somewhat overextended. It is simply important to address the technical concerns and the conceptual issues noted above.

We greatly appreciate the advice and we would like to thank the reviewer for the insightful comments.

REVIEWERS' COMMENTS

Reviewer #1 (Remarks to the Author):

The authors have done a laudable revision. I now support publication with Nature Comms.

Reviewer #2 (Remarks to the Author):

The revised version is fine. I have no more comments.

Reviewer #3 (Remarks to the Author):

The authors have satisfied my concerns. The following are minor comments to correct some terminology.

Minor Comments:

For Fig 1E- while spaw can be bilateral, the other markers are really midline. this is also true for Fig S1C,S3F. Also, for evaluation of the heart position, it is midline and not bilateral (unless the authors are arguing that there is cardiac bifida which i am assuming they are not). spaw and pitx2c are in the lateral margins and so can be bilateral while the heart is midline and fails to loop right or left (just trying to be consistent with the terminology in the LR field).

Also please correct:

“The heart (labeled with lefty2 and mly7), liver, and pancreas (both labeled with foxa3) had a greater frequency of straight and right-sided orientation compared to wt embryos (Fig. 1 E-F). “

In this case “straight” should be midline.

To really test redundancy between *cygb1*, *cyb2*, they need to make double mutants and compare phenotypes. I don't think this is necessary as this is a minor point. I would just remove the part about "compensatory mechanisms" and state that could also give LR phenotypes.

"Injections of *cygb2*-mRNA in *cygb2801a* mutants restored NO/NO-2 to normal levels at early gastrula stage (Fig. 2 E-G) but did not rescue the laterality phenotype observed at 2 dpf, likely due to the mRNA degradation upon injection over the longer period of time necessary to determine correct laterality."

While I appreciate the desire to explain why the rescue doesn't work, this explanation seems unlikely. Indeed, they were able to rescue the *nos2b*-MO phenotype with *nos2b*-mRNA injection which suggests that mRNA rescue is possible for laterality phenotypes. I would simply delete the underlined part of the statement. This might be a good place to state that something like: "while we could not rescue the *cygb2801a* cardiac laterality phenotype with *cygb2* mRNA, we do conclude that our depletion strategies are specific as we have two alleles of *cygb2* that have been bred multiple times, a MO, etc.

REVIEW for Nature Communications of Manuscript Rochon et al. “Cytoglobin regulates NO-dependent cilia motility and organ laterality during development.”

We appreciate the constructive feedback from the reviewers. We provide our answers to their comments below in red.

REVIEWER COMMENTS

Reviewer #1 (Remarks to the Author):

The authors have done a laudable revision. I now support publication with Nature Comms.
We thank the reviewer for the supportive comment.

Reviewer #2 (Remarks to the Author):

The revised version is fine. I have no more comments.
We thank the reviewer for the supportive comment.

Reviewer #3 (Remarks to the Author):

Minor Comments:

For Fig 1E- while spaw can be bilateral, the other markers are really midline. this is also true for Fig S1C,S3F. Also, for evaluation of the heart position, it is midline and not bilateral (unless the authors are arguing that there is cardiac bifida which i am assuming they are not). spaw and pitx2c are in the lateral margins and so can be bilateral while the heart is midline and fails to loop right or left (just trying to be consistent with the terminology in the LR field).
We appreciate the expert comments provided by this reviewer. We have substituted “bilateral” with “midline” in all the figures and text accordingly.

Also please correct:

“The heart (labeled with lefty2 and mly7), liver, and pancreas (both labeled with foxa3) had a greater frequency of straight and right-sided orientation compared to wt embryos (Fig. 1 E-F). “
In this case “straight” should be midline.

We have fixed this accordingly.

To really test redundancy between cygb1, cyb2, they need to make double mutants and compare phenotypes. I don’t think this is necessary as this is a minor point. I would just remove the part about “compensatory mechanisms” and state that could also give LR phenotypes.

We have fixed this accordingly.

“Injections of cygb2-mRNA in cygb2801a mutants restored NO/NO-2 to normal levels at early gastrula stage (Fig. 2 E-G) but did not rescue the laterality phenotype observed at 2 dpf, likely due to the mRNA degradation upon injection over the longer period of time necessary to determine correct laterality.”

While I appreciate the desire to explain why the rescue doesn’t work, this explanation seems unlikely. Indeed, they were able to rescue the nos2b-MO phenotype with nos2b-mRNA injection which suggests that mRNA rescue is possible for laterality phenotypes. I would simply delete the underlined part of the statement. This might be a good place to state that something like: “while we could not rescue the cygb2801a cardiac laterality phenotype with cygb2 mRNA, we do

conclude that our depletion strategies are specific as we have two alleles of *cygb2* that have been bred multiple times, a MO, etc.

The following sentence was added as suggested: “While *cygb2-mRNA* did not rescue the laterality phenotype observed at 2 dpf, we conclude that the depletion strategies used in this study are specific for *Cygb2* due to the reproducibility of the phenotype observed in two mutant alleles and in the morpholino knockdown model. Additionally, outcrossing the mutants over several generations decreases the possibility that Cas9-mediated off target effects underlie the observed phenotype.” (lines 148-152).